# How spin relaxes and dephases in bulk halide perovskites

Junqing Xu[1,2,8], Kejun Li [3,8], Uyen N. Huynh[4], Mayada Fadel [5], Jinsong Huang [6], Ravishankar Sundararaman[5] ✉, Valy Vardeny[4] ✉ & Yuan Ping [3,7] ✉

Spintronics in halide perovskites has drawn significant attention in recent years, due to their highly tunable spin-orbit fields and intriguing interplay with lattice symmetry. Here, we perform first-principles calculations to determine the spin relaxation time ($T_1$) and ensemble spin dephasing time ($T_2^*$) in a prototype halide perovskite, $CsPbBr_3$. To accurately capture spin dephasing in external magnetic fields we determine the Landé $g$-factor from first principles and take it into account in our calculations. These allow us to predict intrinsic spin lifetimes as an upper bound for experiments, identify the dominant spin relaxation pathways, and evaluate the dependence on temperature, external fields, carrier density, and impurities. We find that the Fröhlich interaction that dominates carrier relaxation contributes negligibly to spin relaxation, consistent with the spin-conserving nature of this interaction. Our theoretical approach may lead to new strategies to optimize spin and carrier transport properties.

The field of semiconductor spintronics aims to achieve the next generation of low-power electronics by making use of the spin degree of freedom. Several classes of materials for spintronic applications have been discovered, investigated, and engineered in the past decade[1–5]. Efficient spin generation and manipulation require a large spin-orbit coupling (SOC), with GaAs a prototypical system, whereas long spin lifetimes ($\tau_s$) is mostly found in weak SOC materials, such as graphene and diamond. Materials with large SOC as well as long $\tau_s$ are ideal for spintronic applications but rare, presenting a unique opportunity for the discovery of new materials.

Halide perovskites, known as prominent photovoltaic[6] and light-emitting materials[7] with remarkable optoelectronic properties, have recently attracted interests also for spin-optoelectronic properties[8–13], since these materials exhibit both long lifetimes and large SOC (due to heavy elements). Compared to conventional spintronic materials, the optical accessibility for spin generation and detection of halide perovskites opens up a new avenue for spin-optoelectronics applications.

Additionally, with highly tunable symmetry through the organic-inorganic framework, large Rashba splitting and high spin polarization have been realized at room temperature, critical for device applications. For example, extremely high spin polarization was produced through charge current in chiral nonmagnetic halide perovskites at room temperature in the absence of external magnetic fields[8] ($\mathbf{B}^{ext}$), which is a hallmark in semiconductor spintronics. Persistent spin helix states that preserve SU(2) symmetry and that can potentially provide exceptionally long $\tau_s$ were recently discovered in two-dimensional halide perovskites[12].

Several recent experimental studies have sought to identify the dominant spin relaxation and dephasing mechanisms to further control and elongate $\tau_s$ of halide perovskites[8–11], e.g. via time-resolved Kerr/Faraday rotations. In particular, the bulk halide perovskite such as $CsPbBr_3$, which possesses one of the simplest halide perovskite structures, is a good benchmark system to understand the fundamental physical mechanisms but already presents several outstanding

[1]Department of Physics, Hefei University of Technology, Hefei, Anhui, China. [2]Department of Chemistry and Biochemistry, University of California, Santa Cruz, CA, USA. [3]Department of Physics, University of California, Santa Cruz California, USA. [4]Department of Physics and Astronomy, University of Utah, Salt Lake City, UT, USA. [5]Department of Materials Science and Engineering, Rensselaer Polytechnic Institute, Troy, NY, USA. [6]Department of Applied Physical Sciences, University of North Carolina, Chapel Hill, NC, USA. [7]Department of Materials Science and Engineering, University of Wisconsin-Madison, Madison, WI, USA. [8]These authors contributed equally: Junqing Xu, Kejun Li. ✉e-mail: sundar@rpi.edu; u0027991@utah.edu; yuanping@ucsc.edu

questions. First, what is the intrinsic $\tau_s$ of CsPbBr$_3$? Experimentally this is not possible to isolate due to the unavoidable contributions from defects and nuclear spins. However, the intrinsic $\tau_s$ are essential as the upper limits to guide the experimental optimization of materials. Next, what scattering processes and phonon modes dominate spin relaxation when varying the temperature, carrier density, etc.? This has been extensively studied for carrier relaxation dynamics, but not yet for spin relaxation dynamics. As we show here, the role of electron-phonon (e-ph) coupling, and especially the Fröhlich interaction known to be important for carrier relaxation in halide perovskites[14], can be dramatically different in spin relaxation. Lastly, how do electron and hole $\tau_s$ respond to $\mathbf{B}^{ext}$, and what are the roles of their respective $g$-factor inhomogeneity?[9,10]

To answer these questions, we need theoretical studies of spin relaxation and dephasing due to various scattering processes and SOC, free of experimental or empirical parameters. Previous theoretical work on spin properties of halide perovskites have largely focused on band structure and spin texture[12,15,16], and have not yet addressed spin relaxation and dephasing directly. Here, we apply our recently-developed first-principles real-time density-matrix dynamics (FPDM) approach[17–21], to simulate spin relaxation and dephasing times of free electrons and holes in bulk CsPbBr$_3$. FPDM approach was applied to disparate materials including silicon, (bcc) iron, transition metal dichalcogenides (TMDs), graphene-hBN, GaAs, in good agreement with experiments[17,18,20]. We account for ab initio Landé $g$-factor and magnetic momenta, self-consistent SOC, and quantum descriptions of e-ph, electron-impurity (e-i) and electron-electron (e-e) scatterings. We can therefore reliably predict $\tau_s$ with and without impurities, as a function of temperature, carrier density, and $\mathbf{B}^{ext}$.

## Results and discussions

### Theory

We simulate spin and carrier dynamics based on the FPDM approach[17,18]. We solve the quantum master equation of density matrix $\rho(t)$ in the Schrödinger picture as the following:[18]

$$\frac{d\rho_{12}(t)}{dt} = -\frac{i}{\hbar}\left[H(\mathbf{B}^{ext}),\rho(t)\right]_{12} + \left(\frac{1}{2}\sum_{345}\left\{\begin{array}{c}[I-\rho(t)]_{13}P_{32,45}\rho_{45}(t)\\ -[I-\rho(t)]_{45}P^*_{45,13}\rho_{32}(t)\\ +H.C.\end{array}\right\}\right), \quad (1)$$

where the first and second terms on the right side of Eq. (1) relate to Larmor precession and scattering processes respectively. The scattering processes induce spin relaxation via the SOC. $H(\mathbf{B}^{ext})$ is the electronic Hamiltonian at an external magnetic field $\mathbf{B}^{ext}$. $[H,\rho] = H\rho - \rho H$. H.C. is Hermitian conjugate. The subindex, e.g., "1" is the combined index of $\mathbf{k}$-point and band. $P$ is the generalized scattering-rate matrix considering e-ph, e-i and e-e scattering processes, computed from the corresponding scattering matrix elements and energies of electrons and phonons.

Starting from an initial density matrix $\rho(t_0)$ prepared with a net spin, we evolve $\rho(t)$ through Eq. (1) for a long enough time, typically from hundreds of ps to a few $\mu s$. We then obtain the excess spin observable vector $\delta\mathbf{S}^{tot}(t)$ from $\rho(t)$ (Eq. S1-S2) and extract spin lifetime $\tau_s$ from $\delta\mathbf{S}^{tot}(t)$ using Eq. S3.

Historically, two types of $\tau_s$ - spin relaxation time (or longitudinal time) $T_1$ and ensemble spin dephasing time (or transverse time) $T_2^*$ were used to characterize the decay of spin ensemble or $\delta\mathbf{S}^{tot}(t)$[22,23]. Suppose the spins are initially polarized along $\mathbf{B}^{ext} \neq 0$, if $\delta\mathbf{S}^{tot}(t)$ is measured in the parallel direction of $\mathbf{B}^{ext}$, $\tau_s$ is called $T_1$; if along $\perp\mathbf{B}^{ext}$, it is called $T_2^*$. Note that without considering nuclear spins, magnetic impurities, and quantum interference effects[24], theoretical $\tau_s(\mathbf{B}^{ext}=0)$ should be regarded as $T_1$. See more discussions about spin relaxation/ dephasing in Supporting Information Sec. SI.

Below we first show theoretical results of $T_1$ and $T_2^*$ of bulk (itinerant or delocalized) carriers. For bulk carriers of halide perovskites, $T_1$ are mainly limited by Elliott-Yafet (EY) and D'yakonov-Perel' (DP) mechanisms[25,26]. EY represents the spin relaxation pathway due to mostly spin-flip scattering (activated by SOC). DP is caused by randomized spin precession between adjacent scattering events and is activated by the fluctuation of the SOC fields induced by inversion symmetry broken (ISB). Different from $T_1$, $T_2^*$ is additionally affected by the Landé-$g$-factor fluctuation at transverse $\mathbf{B}^{ext}$. We later generalize our results for other halide perovskites by considering the ISB and composition effects. We at the end discuss $T_2^*$ of localized carriers due to interacting with nuclear spins. By simulating $T_1$ and $T_2^*$, and determining the dominant spin relaxation/dephasing mechanism, we provide answers of critical questions raised earlier and pave the way for optimizing and controlling spin relaxation and dephasing in halide perovskites.

### Spin lifetimes at zero magnetic field

Intrinsic $\tau_s$, free from crystal imperfections and nuclear spin fluctuation, is investigated first, which sets up the ideal limit for experiments. At $\mathbf{B}^{ext} = 0$, bulk CsPbBr$_3$ possesses both time-reversal (nonmagnetic) and spatial inversion symmetries, resulting in Kramers degeneracy of a pair of bands between (pseudo-) up and down spins. Spin relaxation in such systems is conventionally characterized by EY mechanism[26]. To confirm if such mechanism dominates in CsPbBr$_3$, the proportionality between $\tau_s$ and carrier lifetime ($\tau_p$, $\tau_s \propto \tau_p$) is a characteristic signature, as is discussed below. Even for intrinsic $\tau_s$, varying temperature ($T$) and carrier density ($n_c$) would lead to large change, and its trend is informative for mechanistic understanding.

Figure 1 a,b show theoretical $\tau_s$ at $\mathbf{B}^{ext} = 0$, including e-ph and e-e scatterings, as a function of $T$ and $n_c$ respectively, for free electrons and holes (SI Fig. S7). Note that although bulk CsPbBr$_3$ crystal symmetry is orthorhombic, the spin lifetime anisotropy along three principle directions is weak (see SI Fig. S8). Therefore only $\tau_s$ along the [001] direction is presented here. We have several major observations as summarized below.

First, a clear decay of $\tau_s$ as increasing $T$ is observed. As $\tau_s$ with and without e-e scattering (SI Fig. S7) has little difference, this indicates e-ph scattering is the dominant spin relaxation mechanism (without impurities and $\mathbf{B}^{ext}$). Note that with increasing $T$, phonon occupations increase, which enhances the e-ph scattering and thus lowers both carrier ($\tau_p$) and spin ($\tau_s$) lifetime.

Next, $\tau_s$ steeply decreases with increasing $n_c$ at low $T$ but is less sensitive to $n_c$ at high $T$, as shown in Fig. 1b. The trend of $T_1$ decreasing with $n_c$ is consistent with the experimental observation of $T_1$ decreasing with pump power/fluence in halide perovskites[27–30]. At 4 K, $\tau_s$ decreases steeply by three orders of magnitude with $n_c$ increasing from $10^{16}$ cm$^{-3}$ to $10^{19}$ cm$^{-3}$. Such phenomenon was reported previously for monolayer WSe$_2$[18,31], where spin relaxation is dominated by EY mechanism, the same as in CsPbBr$_3$. The cause of such strong $n_c$-dependence at low $T$ is discussed below in more details, attributing to $n_c$ effects on (averaged) spin-flip matrix elements. As a result, at low $T$ and low $n_c$, $\tau_s$ of CsPbBr$_3$ can be rather long, e.g., ~ 200 ns at 10 K and ~ 8 $\mu s$ at 4 K. This is in fact comparable to the ultralong hole $\tau_s$ of TMDs and their heterostructures[18,32,33], ≥2 $\mu s$ at ~ 5 K, again suggesting the advantageous character of halide perovskite in spintronic applications.

Importantly, good agreement between theoretical results and several independent experimental measurements is observed. Our theoretical results agree well with experimental $T_1$ of bulk CsPbBr$_3$[9] (Exp. C) assuming $n_c \approx 10^{18}$ cm$^{-3}$, and CsPbBr$_3$ nanocrystal[34] (Exp. D) assuming $n_c \approx 10^{16}$ cm$^{-3}$, respectively. We further compare theoretical results with our own measured $T_2^*$ (at $B^{ext}$=100 mT; Exp. A). Excellent agreement is observed at $T \gtrsim 10$ K with $n_c$ around $10^{18}$ cm$^{-3}$ (estimated from the experimental averaged pump power). The agreement

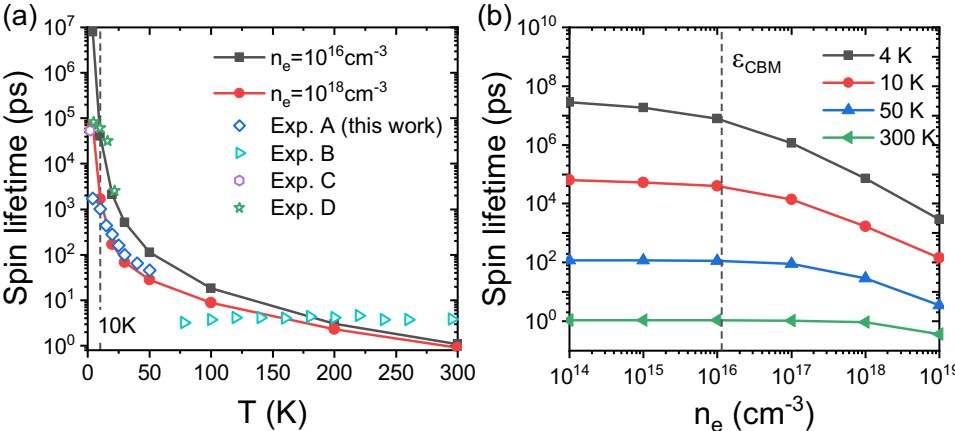

**Fig. 1 | Spin lifetime $\tau_s$ of electrons of CsPbBr$_3$.** We compare electron and hole $\tau_s$ in Supplementary information (SI) Fig. S7 and they have the same order of magnitude at all conditions we investigated. **a** $\tau_s$ due to both the electron-phonon (e-ph) and electron-electron (e-e) scatterings calculated as a function of $T$ at different electron densities $n_e$ compared with experimental data. In Fig. S6, we show $\tau_s$ versus $T$ using log-scale for both $y$- and $x$-axes to highlight low-$T$ region. Exp. A are our experimental data of $T_2^*$ of free electrons in bulk CsPbBr$_3$ at a small external transverse magnetic field. For Exp. A, the density of photo-excited carriers is estimated to be about $10^{18}$ cm$^{-3}$. Exp. B are experimental data of exciton $\tau_s$ of CsPbBr$_3$ films from Ref. 11. Exp. C and Exp. D are experimental data of spin relaxation time $T_1$ of bulk CsPbBr$_3$ and CsPbBr$_3$ nanocrystals measured by the spin inertia method from Ref. 9 and[34] respectively. In Ref. 34, it was declared that quantum confinement effects do not modify the spin relaxation/dephasing significantly (see its Table 1), so that their $T_1$ data can be compared with our theoretical results. For Exp. C and D, the measured lifetimes cannot be unambiguously ascribed to electrons or holes and can be considered as values between electron and hole $T_1$. The carrier densities are not reported for Exp. C and D. **b** $\tau_s$ due to both the e-ph and e-e scatterings as a function of $n_e$ at different $T$. The vertical dashed line in **b** corresponding to $n_e$ with chemical potential $\mu_{F,c}$ at the conduction band minimum (CBM).

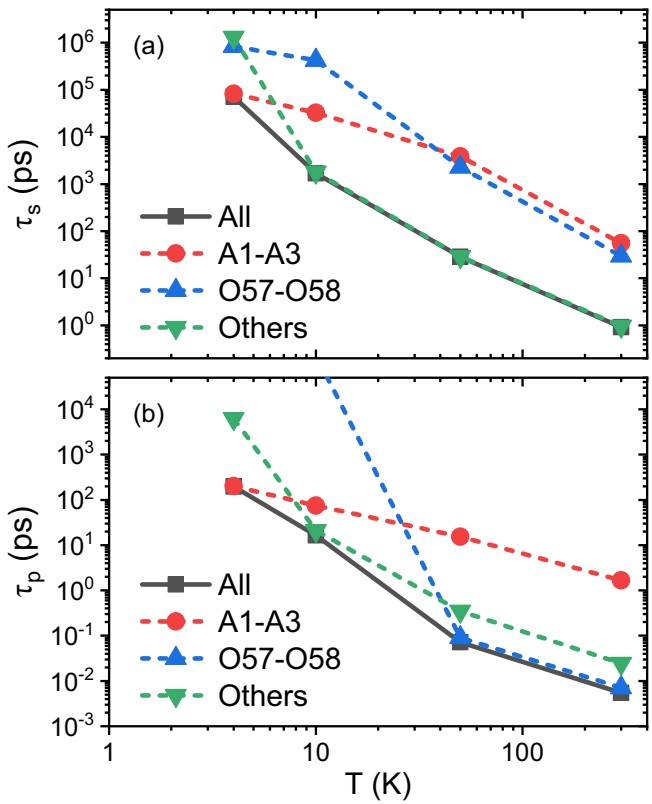

**Fig. 2 | Phonon-mode contribution analysis. a** Spin lifetime $\tau_s$ and **b** carrier lifetime $\tau_p$ due to different phonon modes. "A" and "O" denote acoustic and optical modes respectively. The number index is ordered by increasing phonon energies. The phonon dispersion is given in SI Fig. S2. Here carrier density $n_c$ is set to be $10^{18}$ cm$^{-3}$. We note that special optical phonon modes O57 and O58 are dominant in carrier relaxation above 50 K **b**, consistent with the usual Fröhlich interaction picture, but are not important in spin relaxation **a**.

however becomes worse at $T < 10$ K. The discrepancy is possibly due to nuclear-spin-induced spin dephasing of carriers, as will be discussed in the last subsection.

We then study the effects of the e-i scattering on $\tau_s$ for various point defects. We find that at low $T$, e.g., $T < 20$ K, the e-i scattering reduces $\tau_s$, consistent with EY mechanism (which states increasing extrinsic scatterings reduces spin lifetime). With a high impurity density $n_i$, e.g., $10^{18}$ cm$^{-3}$, the e-i scattering may significantly reduce $\tau_s$ below 10 K, seemingly leading to better agreement between theoretical $\tau_s$ and experimental data from Exp. A, as shown in SI Fig. S9. However, as will be discussed below in the subsection of magnetic-field effects, a relatively high $n_i$ predicts incorrect values of $T_2^*$ and worse agreement with experimental data (Exp. A) on $\mathbf{B}^{\text{ext}}$-dependence. Therefore, the discrepancy between our theoretical $\tau_s$ and our measured $T_2^*$ below 10 K is probably not explained by the impurity scattering effects.

In addition, the electron and hole $\tau_s$ have the same order of magnitude (Fig. S7), consistent with experiments, but in sharp contrast to conventional semiconductors (e.g., silicon and GaAs[35]), which have longer electron $\tau_s$ than hole owing to band structure difference between valence and conduction band edges.

Finally, we also predict the spin diffusion length ($l_s$) of pristine CsPbBr$_3$ in the low-density limit, which sets the upper bound of $l_s$ at different $T$. We use the relation $l_s = \sqrt{D\tau_s}$, where $D$ is diffusion coefficient obtained using the Einstein relation, with carrier mobility $\mu$ from first-principles calculations[19] (more details in Sec. SVII). Excellent agreement between theoretical and experimental carrier mobility is found for CsPbBr$_3$ (SI Fig. S12a). We find $l_s$ is longer than 10 nm at 300 K, and possibly reach tens of $\mu$m at $T \leq 10$ K (see details in Sec. SVII and Fig. S12 in SI).

## Analysis of spin-phonon relaxation

To gain deep mechanistic insights, we next analyze different phonon modes and carrier density effects on spin relaxation through examining spin-resolved e-ph matrix elements.

In Fig. 2, we compare the contribution of different phonon modes to $\tau_s$ and $\tau_p$. First, we find that at a very low $T$ - 4 K, only acoustic modes (A1-A3) contribute to spin and carrier relaxation. This is simply because

the optical phonons are not excited at such low $T$ (corresponding $k_B T \sim 0.34$ meV much lower than optical energy $\gtrsim 2$ meV). At $T \gtrsim 10$ K, optical modes are more important for both spin and carrier relaxation (green and blue dashed lines closer to black line (all phonons) in Fig. 2).

In particular, from Fig. 2b, we find that two special optical modes - 57th and 58th modes (O57-O58, modes ordered by phonon energy with their phonon vector plots in SI Fig. S3) dominate carrier relaxation at $T \gtrsim 50$ K, because $\tau_p$ due to O57-O58 (blue dashed line) nearly overlaps with $\tau_p$ due to all phonon modes (black line). These two optical modes are mixture of longitudinal and transverse vibration as shown in SI Fig. S3. In contrast, for spin relaxation in Fig. 2a, at $T \gtrsim 10$ K O57-O58 are less important than other optical modes (green dashed line). More specifically, in this temperature range, there are tens of phonon modes (with energies ranging from 2 meV to 18 meV), contributing similarly to spin relaxation. This is contradictory to the simple assumption frequently employed in previous experimental studies[9,36,37] that a single longitudinal optical (LO) phonon with a relatively high energy (e.g. ~ 18 meV for CsPbBr$_3$ in Ref. 9) dominates spin relaxation over a wide $T$ range, e.g., from 50 K to 300 K, through a Fröhlich type e-ph interaction.

In the simplified picture of Fermi's golden rule (FGR), $\tau_s^{-1}$ and $\tau_p^{-1}$ (due to e-ph scattering) are proportional to the modulus square of spin-flip ($|\tilde{g}^{\uparrow\downarrow}|^2$) and spin-conserving ($|\tilde{g}^{\uparrow\uparrow}|^2$) matrix elements (ME), respectively. From Fig. 3a, we find that spin-flip ME is dominated by

"other optical modes" (blue line), opposite to the spin-conserving ME in Fig. 3b (i.e. instead, dominated by special optical phonon modes O57 and O58 (red line)). This well explains the different roles of optical O57-O58 modes in carrier and spin relaxation. Moreover, spin-conserving ME for O57-O58 in Fig. 3b diverges at $q \to 0$, which indicates its dominant long-range nature, consistent with the common long-range Fröhlich interaction picture[38], mostly driving carrier relaxation in polar materials at high $T$ (e.g., 300 K). On the contrary, the small magnitude of spin-flip ME for O57-O58 modes indicates that Fröhlich interaction is unimportant for spin relaxation. This is because all spin-dependent parts of the e-ph interaction are short-ranged, while Fröhlich interaction is the only long-range part of the e-ph interaction but is spin-independent. This important conclusion again emphasizes the sharp difference between spin and carrier relaxations in polar materials.

To explain the strong $n_c$ dependence of $\tau_s$ at low $T$, we further analyze the $T$ and chemical potential ($\mu_{F,c}$) dependent effective spin-flip ME $\overline{|\tilde{g}^{\uparrow\downarrow}|^2}$ (averaged around $\mu_{F,c}$, see Eq. (12)) and scattering density of states $D^S$ (Eq. (15)). In FGR, we have the approximate relation in Eq. (16), i.e. $\tau_s^{-1} \propto \overline{|\tilde{g}^{\uparrow\downarrow}|^2} D^S$.

In Fig. 3c, we show the $n_c$ dependence of $\tau_s^{-1}$, compared with $\overline{|\tilde{g}^{\uparrow\downarrow}|^2}$ and $\overline{|\tilde{g}^{\uparrow\downarrow}|^2} D^S$. Indeed we can see $\tau_s^{-1}$ and $\overline{|\tilde{g}^{\uparrow\downarrow}|^2} D^S$ nearly overlapped, as the result of Eq. (16). The strong increase of $\tau_s^{-1}$ at $n_c \gtrsim 10^{16}$ cm$^{-3}$ can be

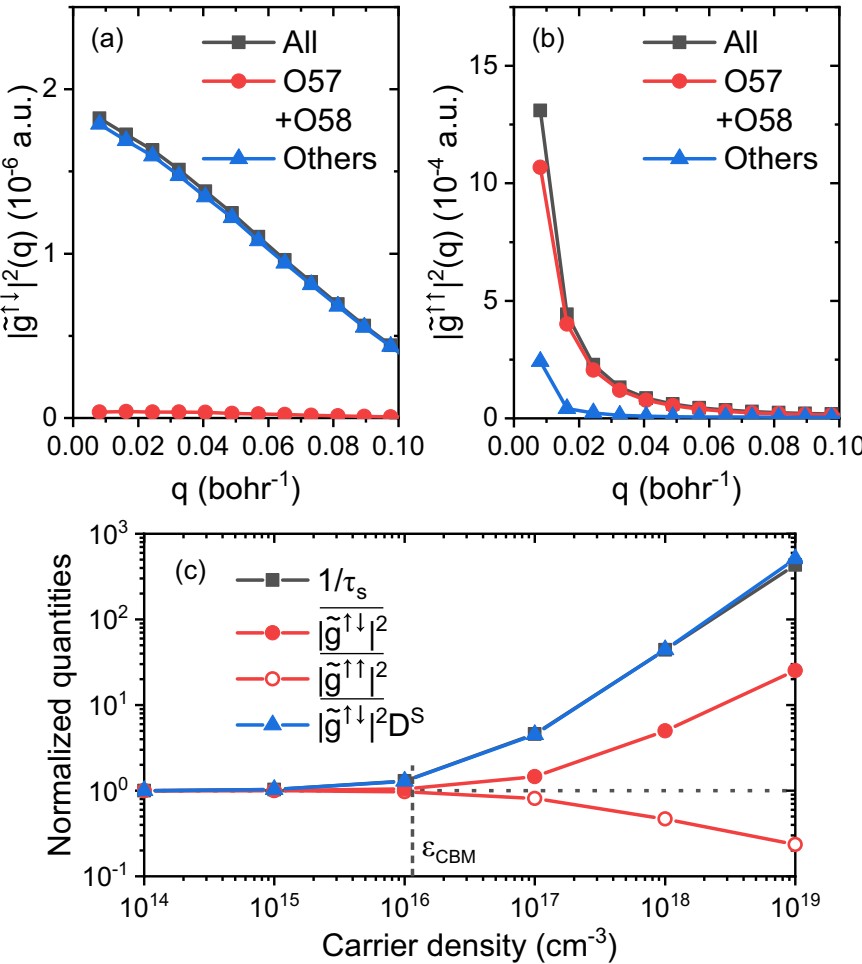

**Fig. 3 | The analysis of the e-ph matrix elements (ME). a** The $q$-resolved modulus square of spin-flip e-ph ME $\overline{|\tilde{g}^{\uparrow\downarrow}|^2}(q)$ at a high temperature - 300 K with a part of or all phonon modes. **b** The same as **a** but for spin-conserving e-ph ME $\overline{|\tilde{g}^{\uparrow\uparrow}|^2}(q)$. **c** $\overline{|\tilde{g}^{\uparrow\downarrow}|^2}$, $\overline{|\tilde{g}^{\uparrow\uparrow}|^2}$ and $\overline{|\tilde{g}^{\uparrow\downarrow}|^2} D^S$ of conduction electrons as a function of carrier density at a low $T$ - 10 K compared with the spin relaxation rates $1/\tau_s$. $\overline{|\tilde{g}^{\uparrow\downarrow}|^2}$ and $\overline{|\tilde{g}^{\uparrow\uparrow}|^2}$ are the $T$ and $\mu_{F,c}$ dependent effective (averaged around the band edge or $\mu_{F,c}$) modulus square of spin-flip and spin-conserving e-ph ME, respectively (see Eq. (12)). $D^S$ is the scattering density of states (Eq. (15)). The vertical dashed line corresponding to $\mu_{F,c}$ at CBM.

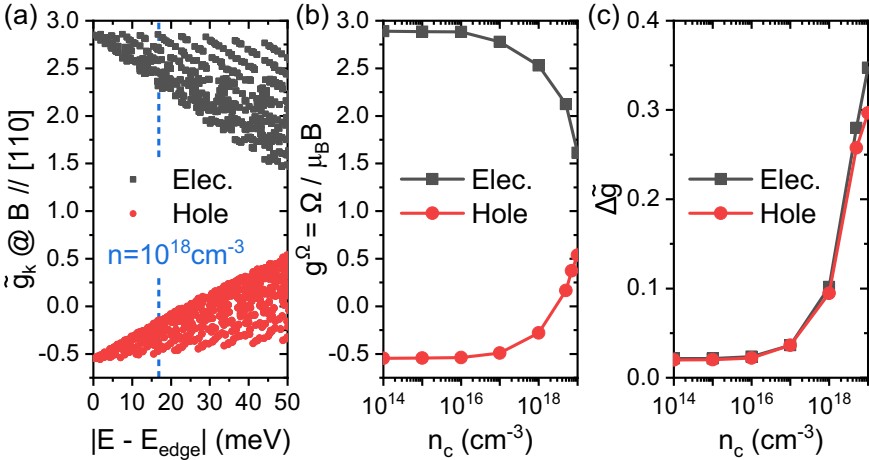

**Fig. 4 | The Landé g-factors of electrons and holes calculated at the PBE functional.** The external magnetic fields $\mathbf{B}^{\text{ext}}$ are along [110] direction. **a** The **k**-dependent g-factor $\widetilde{g}_k$ (Eq. (8) and (9)) at **k** points around the band edges. Each data-point corresponds to a **k** point. **b** The global g-factor $g^\Omega = \Omega/\mu_B B$ as a function of $n_c$, where $\Omega$ is Larmor precession frequency extracted from spin dynamics at $\mathbf{B}^{\text{ext}} \neq 0$. $g^\Omega = \pm |g^\Omega|$ if the excess/excited spin $\delta \mathbf{S}^{\text{tot}}(t)$ precesses along $\pm \delta \mathbf{S}^{\text{tot}}(t) \times \mathbf{B}^{\text{ext}}$. $g^\Omega$ is close to the averaged g-factor $\overline{\widetilde{g}}$ defined in Eq. (10). **c** The effective amplitude of the fluctuation of g factors - $\Delta\widetilde{g}$ defined in Eq. (11) as a function of carrier density at 10 K.

attributed to the fact that both spin-flip ME $\overline{|\widetilde{g}^{\uparrow\downarrow}|^2}$ and scattering density of states $D^s$ increase with $n_c$. Interestingly, the effective spin-conserving ME $\overline{|\widetilde{g}^{\uparrow\uparrow}|^2}$, most important in carrier relaxation, decreases with $n_c$, opposite to spin-flip $\overline{|\widetilde{g}^{\uparrow\downarrow}|^2}$. This again emphasizes the e-ph scattering affects carrier and spin relaxation differently, given the opposite trends of spin-conserving and spin-flip scattering as a function of $n_c$. When $n_c < 10^{16}$ cm$^{-3}$, $\tau_s^{-1}$ is insensitive to $n_c$, which is because both $\overline{|\widetilde{g}^{\uparrow\downarrow}|^2}$ and $D^s$ are determined by e-ph transitions around the band edge. In "Methods" section, we have proven that at the low density limit, since carrier occupation satisfies Boltzmann distribution, both $\overline{|\widetilde{g}^{\uparrow\downarrow}|^2}$ and $D^s$ are $\mu_{F,c}$ and $n_c$ independent.

### Landé g-factor and transverse-magnetic-field effects

At $\mathbf{B}^{\text{ext}}$, the electronic Hamiltonian reads

$$H_k(\mathbf{B}^{\text{ext}}) = H_{0,k} + \mu_B \mathbf{B}^{\text{ext}} \cdot (\mathbf{L}_k + g_0 \mathbf{S}_k), \tag{2}$$

where $\mu_B$ is Bohr magneton; $g_0$ is the free-electron g-factor; $\mathbf{S}$ and $\mathbf{L}$ are the spin and orbital angular momentum respectively. The simulation of $\mathbf{L}$ is nontrivial for periodic systems and the details are given in Method section and Ref. 39. Having $H(\mathbf{B}^{\text{ext}})$ at a transverse $\mathbf{B}^{\text{ext}}$ perpendicular to spin direction, $T_2^*$ is obtained by solving the density-matrix master equation in Eq. (1).

The key parameters for the description of the magnetic-field effects are the Landé g-factors. Their values relate to $\mathbf{B}^{\text{ext}}$-induced energy splitting (Zeeman effect) $\Delta E_k(\mathbf{B}^{\text{ext}})$ and Larmor precession frequency $\Omega_k$, satisfying $\Omega_k \approx \Delta E_k = \mu_B B^{\text{ext}} \widetilde{g}_k$ with $\widetilde{g}_k$ the k-resolved Landé g-factor. More importantly, the g-factor fluctuation (near Fermi surface or $\mu_{F,c}$) leads to spin dephasing at transverse $\mathbf{B}^{\text{ext}}$, corresponding to $T_2^*$.

Figure 4a shows $\widetilde{g}_k$ of electrons and holes at **k**-points around the band edges. $\widetilde{g}_k$ are computed using $\Delta E_k(\mathbf{B}^{\text{ext}})$ (Eq. (8) and (9)) obtained from $H_k(\mathbf{B}^{\text{ext}})$. Our calculated electron $\widetilde{g}_k$ are larger than hole $\widetilde{g}_k$, and the sum of electron and hole $\widetilde{g}_k$ range from 1.85 to 2.4, in agreement with experiments[9,35]. Furthermore, $\widetilde{g}_k$ are found sensitive to state energies and wavevectors **k**, and the fluctuation of $\widetilde{g}_k$ is enhanced with increasing the state energy. In Fig. 4b,c, we show the global g-factor $g^\Omega$ and the amplitude of the g-factor fluctuation (near the Fermi surface) $\Delta\widetilde{g}$ (Eq. (11)) as a function of $n_c$. Both $g^\Omega$ and $\Delta\widetilde{g}$ are insensitive to $n_c$ at $n_c < 10^{16}$ cm$^{-3}$, but sensitive to $n_c$ at $n_c \gtrsim 10^{16}$ cm$^{-3}$.

In Fig. 4, we show ab initio g-factors computed with the PBE functional[40]. We further compare g-factors computed using different exchange-correlation functionals ($V_{xc}$) in SI Sec. SV. It is found that the magnitude of $\Delta\widetilde{g}$ and the trend of g-factor change with the state energy are both insensitive to $V_{xc}$. Since $T_2^*$ only depends on $\Delta\widetilde{g}$, our predictions of $T_2^*$ should be reliable.

Next, we discuss magnetic-field effects on $\tau_s$ in Fig. 5, calculated from our FPDM approach, and analyze them with phenomenological models. At transverse $\mathbf{B}^{\text{ext}}$, the total spin decay rate is approximately expressed by

$$\tau_s^{-1}(\mathbf{B}^{\text{ext}}) \approx (\tau_s^0)^{-1} + (\tau_s^{\Delta\Omega})^{-1}(\mathbf{B}^{\text{ext}}), \tag{3}$$

where $(\tau_s^0)^{-1}$ is the zero-field spin relaxation rate due to EY mechanism; $(\tau_s^{\Delta\Omega})^{-1}$ is induced by the Larmor-precession-frequency fluctuation ($\Delta\Omega = \mu_B B^{\text{ext}} \Delta\widetilde{g}$), and can be described by different mechanisms depending on the magnitude of $\tau_p \Delta\Omega$[22,26]:

(i) Free induction decay (FID) mechanism if $\tau_p \Delta\Omega \gtrsim 1$ (weak scattering limit). We have

$$(\tau_s^{\Delta\Omega})^{-1} \sim (\tau_s^{\text{FID}})^{-1} \sim C^{\Delta g} \Delta\Omega = C^{\Delta g} \mu_B B^{\text{ext}} \Delta\widetilde{g}, \tag{4}$$

where $C^{\Delta g}$ is a constant and often taken as 1 or $1/\sqrt{2} \approx 0.71$[9,22,36,41–44]. The latter assumes a Gaussian distribution of g-factors and the scattering being completely absent[22,42,44].

(ii) Dyakonov Perel (DP) mechanism if $\tau_p \Delta\Omega \ll 1$ (strong scattering limit). We have

$$(\tau_s^{\Delta\Omega})^{-1} \sim (\tau_s^{\text{DP}})^{-1} \sim \tau_p (\Delta\Omega)^2 = \tau_p (\mu_B B^{\text{ext}} \Delta\widetilde{g})^2. \tag{5}$$

(iii) Between (i) and (ii) regimes, there isn't a good approximate relation for $(\tau_s^{\Delta\Omega})^{-1}$, but we may expect that[22]

$$(\tau_s^{\text{DP}})^{-1} < (\tau_s^{\Delta\Omega})^{-1} < (\tau_s^{\text{FID}})^{-1}. \tag{6}$$

From Fig. 5a, we find that magnetic-field effects are weak ($\tau_s(\mathbf{B}^{\text{ext}})/\tau_s(0) \approx 1$) at $T \gtrsim 20$ K. This is because at high $T$, e-ph scattering is strong which leads to short $\tau_p$ and short spin lifetime at zero field $\tau_s^0$ (large $(\tau_s^0)^{-1}$). Then the spin relaxation falls into strong or intermediate scattering regimes ((ii) or (iii)), which give small $(\tau_s^{\Delta\Omega})^{-1}$. Finally, following $(\tau_s^{\Delta\Omega})^{-1} \ll (\tau_s^0)^{-1}$ obtained above, we reach $\tau_s(\mathbf{B}^{\text{ext}})/\tau_s(0) \approx 1$ from Eq. (3).

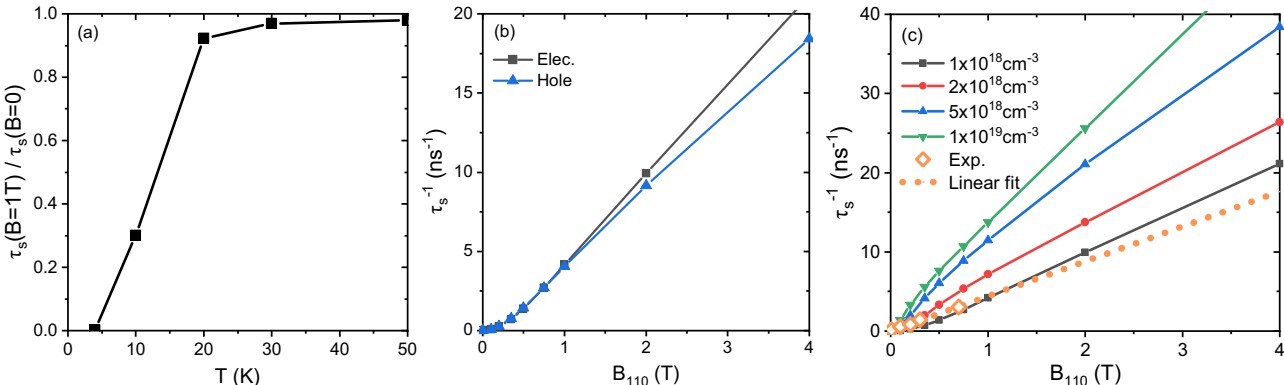

**Fig. 5 | The effects of transverse $B^{ext}$ (perpendicular to spin direction) on calculated $\tau_s$ of free carriers of CsPbBr$_3$. a** The ratio of $\tau_s$ at $B^{ext}$=1 T and $\tau_s$ at $B^{ext}$=0 as a function of $T$. Here electron carrier density $n_e$=10$^{18}$ cm$^{-3}$. **b** Spin decay rates ($\tau_s^{-1}$) of electrons and holes as a function of $B^{ext}$ at 4 K with $n_c$ = 10$^{18}$ cm$^{-3}$. **c** $\tau_s^{-1}$ as a function of $B^{ext}$ at 4 K at different $n_e$. "Exp." (orange open diamond) represent our experimental data (with $B^{ext}$ along [010] direction), where the density of photo-excited carriers is estimated about 10$^{18}$ cm$^{-3}$. The orange dashed line is the linear fit of experimental data. The linear relation between ensemble spin dephasing rate and $B^{ext}$ was frequently found and used in previous experimental measurements[9,36,43,44].

As a result, we only discuss $\tau_s$ at $B^{ext} \neq 0$ below 20 K, specifically at 4K afterwards. From Fig. 5b, we can see that magnetic-field effects on electron and hole $\tau_s$ are quite similar, which is a result of their similar band curvatures, e-ph scattering, and $\Delta\tilde{g}$ (Fig. 4c), although their absolute $g$-factors are quite different, as shown in Fig. 4a,b.

We further examine magnetic-field effects on $\tau_s$ at 4 K in Fig. 5c. As discussed above, $\tau_s^{-1}(B^{ext})$ increases with $B^{ext}$. More specifically, we find that the calculated $\tau_s^{-1}(B^{ext})$ is proportional to $(B^{ext})^2$ at low $B^{ext}$ (details in SI Fig. S13) following the DP mechanism (Eq. (5)), but linear to $B$ at higher $B$ following the free induction decay mechanism(Eq. (4)).

The comparison of calculated $\tau_s^{-1}(B^{ext})$ with experimental data (orange diamond in Fig. 5b) is reasonable with $n_e$ around 10$^{18}$ cm$^{-3}$ (the experimental estimated average $n_c$). However, their $B^{ext}$-dependence is not the same in the small $B^{ext}$ range, e.g. at $B^{ext}$<0.4 Tesla, the calculated $\tau_s^{-1}(B^{ext})$ is proportional to $(B^{ext})^2$ (as shown in SI Fig. S13), whereas the experimental $\tau_s^{-1}$ is more likely linear to $B^{ext}$. In principles, extremely small $B^{ext}$ will lead to $\Delta\Omega$ small enough falling in the DP regime $((\tau_s^{\Delta\Omega})^{-1}$ proportional to $(B^{ext})^2)$. However, experimental results still keep in the FID regime $((\tau_s^{\Delta\Omega})^{-1}$ linear dependent on $B^{ext})$ at small $B^{ext}$. This inconsistency implies additional magnetic field fluctuation contributes to $\Delta\Omega$ and/or other faster spin dephasing processes exist at small external $B^{ext}$. It may originate from nuclear spin fluctuation, magnetic impurities, carrier localization, chemical potential fluctuation, etc.[9,35] in samples, which are however rather complicated for a fully first-principles description. In this work, we focus on spin dephasing of bulk carriers due to Zeeman effects and various scattering processes.

Moreover, in Fig. 5c, we find that at $B^{ext} \neq 0$, $\tau_s$ decreases with $n_c$, similar to the case at $B^{ext} = 0$. But the origin of the strong $n_c$ dependence at high $B^{ext}$ is very different from $\tau_s$ at $B^{ext} = 0$. When $B^{ext} \geq 0.4$ Tesla, $\tau_s$ is dominated by the FID mechanism (Eq. (4)), thus its $n_c$ dependence is mostly from $\Delta\tilde{g}$'s strong $n_c$ dependence shown in Fig. 4c.

Finally, we show $\tau_s^{-1}(B^{ext})$ as a function of $B^{ext}$ at 4 K with the e-i scattering in Fig. S14. We find that with relatively strong impurity scattering (e.g, with 10$^{17}$ cm$^{-3}$ V$_{Pb}$ neutral impurities), the $B^{ext}$-dependence of $\tau_s$ becomes quite weak, in disagreement with experiments, indicating that impurity scattering is probably weaker in those experiments. See more discussions in Sec. SVIII.

### Inversion symmetry broken (ISB), composition effects and hyperfine coupling

For halide perovskites, ISB may present due to ferroelectric polarization, strain, surface, applying electric fields, etc. One of the most important effects from ISB is inducing **k**-dependent SOC fields (called $\mathbf{B}^{in}$). $\mathbf{B}^{in}$ can change the electronic energies and spin textures, which may significantly modify the spin relaxation/dephasing. To understand the ISB effects, we simulate $\tau_s$ with two important types of $\mathbf{B}^{in}$ - Rashba and PSH (persistent spin helix) ones. Rashba SOC presents in many 2D and 3D materials, e.g., wurtzite GaN and graphene on SiO$_2$/hBN. PSH exhibits SU(2) symmetry[45,46] which is robust against spin-conserving scattering, and was recently realized in 2D hybrid perovskites[12]. Their effects are considered by introducing an additional SOC term to the electronic Hamiltonian perturbatively. The specific forms of Rashba and PSH SOC Hamiltonians are given in Eq. (18)-(22) in "Methods" section.

From Fig. 6a, we find that $\tau_s$ is reduced by Rashba SOC and the reduction is significant when the SOC coefficient $\alpha \geq 0.5$ eVÅ. This is because Rashba SOC induces a nonzero $\Delta\Omega \propto \alpha$ and then induces an DP/FID spin decay channel additional to the EY one. Similar to Eq. (3), the total rate $\tau_s^{-1} \approx \tau_s^{-1}|_{\alpha=0} + (\tau_s^{-1})^{\Delta\Omega}$. At $\alpha \geq 0.5$ eVÅ, $(\tau_s^{-1})^{\Delta\Omega}$ becomes large, so that $\tau_s$ is significantly reduced from $\tau_s^{-1}|_{\alpha=0}$. $\tau_s$ keeps decreasing with $\alpha$ but its low limit is bound by $\tau_p$. On the other hand, with PSH SOC, $\tau_s$ (along PSH $\mathbf{B}^{in}$ - $\mathbf{B}^{PSH}$, which is along $y$ direction here) is unchanged at $\alpha \leq 2$ eVÅ, and increases at a larger $\alpha$. The reason is: with PSH SOC, spins are highly polarized along $\mathbf{B}^{PSH}$, so that $\tau_s$ along $\mathbf{B}^{PSH}$ is still dominated by EY mechanism (no spin precession). One critical effect of $\mathbf{B}^{PSH}$ is then modifying the energies (spin split energies). At small $\alpha$, the energy changes are not significant compared with $k_B T$, so that the e-ph scattering contribution to spin relaxation is not modified much; as a result, $\tau_s$ is close to $\tau_s|_{\alpha=0}$. From Fig. 6b, we can see that at large $\alpha$ (e.g., 7 eVÅ) the band structure is however significantly changed. The valence band maxima are now at two opposite $k$-points away from Γ and with opposite spins. Therefore, at large $\alpha$, spin relaxation is dominated by the spin-flip scattering processes between two opposite valleys away from Γ. This can lead to longer $\tau_s$ since the spin-flip processes within one valley (intravalley scattering) are suppressed, essentially a spin-valley locking condition is realized[12,19]. Our FPDM simulations with model SOC suggest that Rashba SOC likely reduces $\tau_s$ while PSH SOC can enhance $\tau_s$ as anticipated in previous experimental study[46]. Note that in practical materials, the ISB effects may not be completely captured by model SOC fields as introduced here. Although in general, we include self-consistent SOC in our FPDM calculations instead of perturbatively, but since the studied equilibrium bulk structure has inversion symmetry, we therefore have to include model ISB SOC perturbatively to simulate such effects induced by various causes. Therefore, further FPDM simulations of materials with ISB structures are important for comprehensive understanding of the

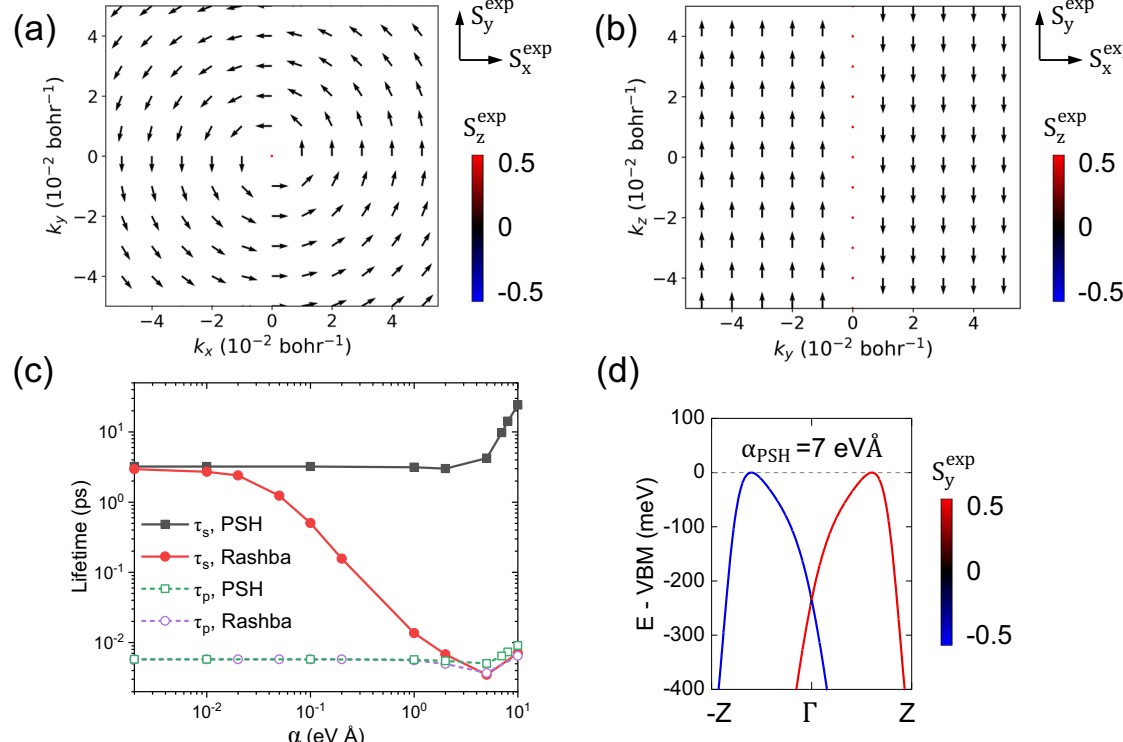

**Fig. 6 | The effects of model SOC fields. a** Spin textures in the $k_x - k_y$ plane of the CsPbBr$_3$ system with model Rashba SOC. $\mathbf{S}^{\text{exp}} \equiv \left( S_x^{\text{exp}}, S_y^{\text{exp}}, S_z^{\text{exp}} \right)$ with $S_i^{\text{exp}}$ being spin expectation value along direction $i$ and is the diagonal element of spin matrix $s_i$ in Bloch basis. The arrow represents the spin orientation in the $S_x^{\text{exp}} - S_y^{\text{exp}}$ plane. The color scales $S_z^{\text{exp}}$. **b** Spin textures in the $k_y - k_z$ plane of the CsPbBr$_3$ system with model PSH (persistent spin helix) SOC. **c** Spin lifetime $\tau_s$ and carrier lifetime $\tau_p$ of CsPbBr$_3$ holes at 300 K considering the effects of Rashba or PSH SOC. $\alpha$ is the Rashba/PSH SOC strength coefficient. Rashba fields have spin texture perpendicular to **k** direction, in the same plane ($xy$ plane here) surrounding $\Gamma$ point. PSH fields have spin texture parallel along the same axis ($y$ axis here). The detailed forms of the model SOC fields and Hamiltonians are given in Eqs. (18)-(22) in "Methods" section. $\tau_s$ is perpendicular to the SOC-field plane for Rashba SOC and is along the high-spin-polarization axis for PSH SOC respectively. **d** The band structure of valence bands considering PSH SOC with $\alpha$=7 eVÅ. The color scales the $S_y^{\text{exp}}$ in **d**.

ISB effects. From Fig. 6a, we find that $\tau_s$ is reduced by Rashba SOC and the reduction is significant when the SOC coefficient $\alpha \geq 0.5$ eVÅ. This is because Rashba SOC induces a nonzero $\Delta\Omega \propto \alpha$ and then induces an DP/FID spin decay channel additional to the EY one. Similar to Eq. (3), the total rate $\tau_s^{-1} \approx \tau_s^{-1}|_{\alpha=0} + \left( \tau_s^{-1} \right)^{\Delta\Omega}$. At $\alpha \geq 0.5$ eVÅ, $\left( \tau_s^{-1} \right)^{\Delta\Omega}$ becomes large, so that $\tau_s$ is significantly reduced from $\tau_s^{-1}|_{\alpha=0}$. $\tau_s$ keeps decreasing with $\alpha$ but its low limit is bound by $\tau_p$. On the other hand, with PSH SOC, $\tau_s$ (along PSH $\mathbf{B}^{\text{in}}$ - $\mathbf{B}^{\text{PSH}}$, which is along $y$ direction here) is unchanged at $\alpha \leq 2$ eVÅ, and increases at a larger $\alpha$. The reason is: with PSH SOC, spins are highly polarized along $\mathbf{B}^{\text{PSH}}$, so that $\tau_s$ along $\mathbf{B}^{\text{PSH}}$ is still dominated by EY mechanism (no spin precession). One critical effect of $\mathbf{B}^{\text{PSH}}$ is then modifying the energies (spin split energies). At small $\alpha$, the energy changes are not significant compared with $k_B T$, so that the e-ph scattering contribution to spin relaxation is not modified much; as a result, $\tau_s$ is close to $\tau_s|_{\alpha=0}$. From Fig. 6b, we can see that at large $\alpha$ (e.g., 7 eVÅ) the band structure is however significantly changed. The valence band maxima are now at two opposite $k$-points away from $\Gamma$ and with opposite spins. Therefore, at large $\alpha$, spin relaxation is dominated by the spin-flip scattering processes between two opposite valleys away from $\Gamma$. This can lead to longer $\tau_s$ since the spin-flip processes within one valley (intravalley scattering) are suppressed, essentially a spin-valley locking condition is realized[12,19]. Our FPDM simulations with model SOC suggest that Rashba SOC likely reduces $\tau_s$ while PSH SOC can enhance $\tau_s$ as anticipated in previous experimental study[46]. Note that in practical materials, the ISB effects may not be completely captured by model SOC fields as introduced here. Although in general, we include self-consistent SOC in our FPDM calculations instead of perturbatively, but since the studied equilibrium bulk structure has inversion symmetry, we therefore have to include model ISB SOC perturbatively to simulate such effects induced by various causes. Therefore, further FPDM simulations of materials with ISB structures are important for comprehensive understanding of the ISB effects.

Furthermore, it is crucial to understand the chemical composition effects to improve our understandings of spin dynamics and transport in many other kinds of halide perovskites beside CsPbBr$_3$. As an initial study, we performed FPDM simulations of $\tau_s$ of holes of pristine bulk CsPbCl$_3$, CsPbI$_3$, MAPbBr$_3$ and CsSnBr$_3$ as a function of temperature, at the same carrier density. We consider the inversion-symmetric orthorhombic phase for all systems, the same as CsPbBr$_3$ here, in order to study chemical composition effect alone. From Fig. 7, our FPDM simulations show that the differences of $\tau_s$ of CsPbBr$_3$, CsPbCl$_3$, CsPbI$_3$, MAPbBr$_3$ and CsSnBr$_3$ are mostly tens of percent or a few times in the wide temperature range from 4 K to 300 K. Specifically, $\tau_s$ of MAPbBr$_3$ is found always shorter than CsPbBr$_3$. $\tau_s$ of CsSnBr$_3$ is found slightly longer than CsPbBr$_3$ at 300 K but becomes shorter than CsPbBr$_3$ at $T$<100 K. A trend of hole $\tau_s$ is found for CsPbX$_3$: $\tau_s$(CsPbCl$_3$) > $\tau_s$(CsPbBr$_3$) > $\tau_s$(CsPbI$_3$), indicating that the lighter the halogen atom, the longer the spin lifetime. This trend may be partly due to two reasons: (i) For the band gap, we have CsPbCl$_3$ > CsPbBr$_3$ > CsPbI$_3$ (1.40, 1.03, and 0.75 eV respectively at PBE), so that spin mixing due to the conduction-valence band mixing is reduced at lighter halogen compound, which usually weakens the spin-phonon interaction; (ii) The lighter halogen atom reduces the SOC strength of the material (weaker SOC reduces the spin mixing between up and down states). Additionally, we find that for all these inversion-symmetric orthorhombic

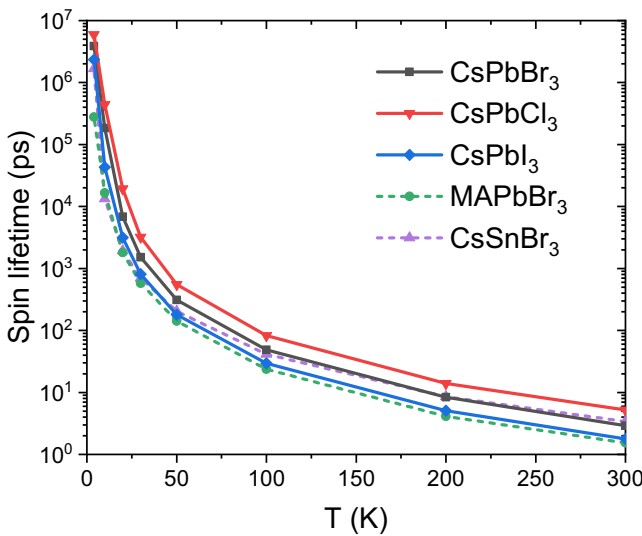

**Fig. 7 | Spin lifetime dependence on chemical compositions.** Spin lifetimes of holes of bulk $CsPbBr_3$, $CsPbCl_3$, $CsPbI_3$, $MAPbBr_3$, and $CsSnBr_3$ as a function of temperature with carrier density $10^{16}$ cm$^{-3}$.

materials, the anisotropy of $\tau_s$ along different crystalline directions is rather weak (see SI Fig. S8). Overall, our results indicate that the chemical composition effects on $\tau_s$ are not very strong when comparing with the effects of the symmetry change (e.g. broken inversion symmetry resulting in Rashba or PSH discussed in Fig. 6). A more systematic study of the composition, symmetry, and dimensionality effects is of great importance and will be our future work.

Above we focus on spin relaxation/dephasing of bulk (or itinerant or delocalized) electrons, for which hyperfine coupling is usually unimportant[22,47]. In actual samples, due to polarons, ionized impurities, etc., a considerable portion of electron carriers are localized. It is known that hyperfine coupling can induce spin dephasing of localized electrons through spin precessions about randomly-distributed nuclear-spin (magnetic) fields $\mathbf{B}^{Nuclear}$. When nuclear spins are weakly polarized (because of weak $\mathbf{B}^{ext}$), $T_2^*$ of localized electrons - $T_{2,loc}^*$ is often estimated based on FID mechanism $1/T_{2,loc}^* \sim \sigma_{\Omega_N}$[48–50], where $\Omega_N$ is Larmor frequency of a localized electron due to $\mathbf{B}^{Nuclear}$ and $\sigma_{\Omega_N}$ is the parameter describing its fluctuation or determining its distribution (Eq. (28) for $\mathbf{B}^{ext}$=0). According to Refs. 9,48–50, $\sigma_{\Omega_N}^2 \sim C^{loc}/V^{loc}$ (Eq. (30)), where $V^{loc}$ is the localization volume. At $\mathbf{B}^{ext}$=0, $C^{loc}$ is determined by hyperfine constant $A$, nuclear spin $I$, isotope abundance and unitcell volume (Eq. (31)). See detailed formulae and our estimates of the above quantities in the "Methods" section. Our estimated $C^{loc}$ is ~ 180 and ~ 530 nm$^3$ ns$^{-2}$ for electrons and holes respectively. The estimated localization radii for halide perovskites are 2.5–14 nm[51–54], which lead to $T_{2,loc}^*(\mathbf{B}^{ext} = 0)$ ~ 0.6–8.0 ns for electrons and ~ 0.35-4.6 ns for holes. Since bulk and localized carriers coexist in materials, $T_{2,loc}^*$ roughly gives the lower bound of the effective carrier $T_2^*$.

In addition to the hyperfine coupling for spin dephasing of localized carriers above, the fluctuation of hyperfine coupling for bulk (delocalized) carriers at different $k$-points may lead to spin dephasing when nuclear spins are polarized along a non-zero transverse $\mathbf{B}^{ext}$. This effect is however rather complicated (probably requiring the difficult $\mathbf{L}$ contribution[55] to hyperfine coupling), beyond the scope of this work.

In summary, through a combined ab initio theory and experimental study, we reveal the spin relaxation and dephasing mechanism of carriers in halide perovskites. Using our FPDM approach and implementing ab initio magnetic momenta and $g$-factor, we simulate free-carrier $\tau_s$ as a function of $T$ and $\mathbf{B}^{ext}$, in excellent agreement with experiments. The transverse magnetic-field effects are found only significant at $T<20$ K. We predict ultralong $T_1$ of pristine $CsPbBr_3$ at low

$T$, e.g., ~ 200 ns at 10 K and ~ 8 $\mu$s at 4 K. We find strong $n_c$ dependence of both $T_1$ and $T_2$ at low $T$, e.g. $\tau_s$ can be tuned by three order of magnitude with $n_c$ from the low density limit to $10^{19}$ cm$^{-3}$. The reasons are attributed to the strong electronic-energy-dependences of spin-flip e-ph matrix elements and $\Delta\tilde{g}$ for $T_1$ and $T_2^*$ respectively. From the analysis of e-ph matrix elements, we find that contrary to common belief, Fröhlich interaction is unimportant to spin relaxation, although critical for carrier relaxation. We further study ISB and composition effects on $\tau_s$ of halide perovskites. We find that ISB effects can significantly change $\tau_s$, i.e. spin lifetime can increase with PSH SOC, but not with Rashba SOC. The composition effect is found not very strong and only changes $\tau_s$ by tens of percent or a few times in a wide temperature. Our work provides fundamental insights on how to control and manipulate spin relaxation in halide perovskites, which are vital for their spintronics and quantum information applications.

## Methods
### Spin dynamics and transport
Spin dynamics and spin lifetime $\tau_s$ are simulated by our recently developed first-principles density-matrix dynamics (FPDM) method[17–21]. Starting from an initial state with a spin imbalance, we evolve the time-dependent density matrix $\rho(t)$ through the quantum master equation with Lindblad dynamics for a long enough simulation time, typically from ns to $\mu$s, varying with systems. After obtaining the excess spin observable $\delta\mathbf{S}^{tot}(t)$ from $\rho(t)$ and fitting $\delta\mathbf{S}^{tot}(t)$ to an exponentially oscillating decay curve, the decay constant $\tau_s$ and the precession frequency $\Omega$ are then obtained (Eq. S3 and Fig. S1 in SI). All required quantities of FPDM simulations, including electron energies, phonon eigensystems, e-ph and e-i scattering matrix elements, are calculated on coarse $k$ and $q$ meshes using the DFT open source software JDFTx[56], and then interpolated to fine meshes in the basis of maximally localized Wannier functions[57–59]. The e-e scattering matrix is computed using the same method given in Ref. 18. More theoretical background and technical details are given in Ref. 19 and[18], as well as the Supporting Information.

Using the same first-principles electron and phonon energies and matrix elements on fine meshes, we calculate the carrier mobility by solving the linearized Boltzmann equation using a full-band relaxation-time approximation[60] and further estimate spin diffusion length based on the drift-diffusion model (SI Sec. SVII).

### Orbital angular momentum
With the Blöch basis, the orbital angular momentum reads

$$\mathbf{L}_{k,mn} = i\left\langle \frac{\partial u_{km}}{\partial \mathbf{k}} \left| \times \left( \widehat{H}_0 - \frac{\epsilon_{km} + \epsilon_{kn}}{2} \right) \right| \frac{\partial u_{kn}}{\partial \mathbf{k}} \right\rangle, \quad (7)$$

where $m$ and $n$ are band indices; $\epsilon$ and $u$ are electronic energy and the periodic part of the wavefunction, respectively; $\widehat{H}_0$ is the zero-field Hamiltonian operator. Eq. (7) can be proven equivalent to $\mathbf{L} = 0.5^*(\mathbf{r} \times \mathbf{p} - \mathbf{p} \times \mathbf{r})$ with $\mathbf{r}$ and $\mathbf{p}$ the position and momentum operator respectively. The detailed implementation of Eq. (7) is described in Ref. 39. Our implementation of $\mathbf{L}$ has been benchmarked against previous theoretical and experimental data for monolayer $MoS_2$ (Table S1).

### $g$-factor of free carriers
In experimental and model Hamiltonian theory studies[9,35], $g$-factor is defined from the ratio between either $\mathbf{B}^{ext}$-induced energy splitting $\Delta E(\mathbf{B}^{ext})$ or Larmor precession frequency $\Omega(\mathbf{B}^{ext})$ to $\mu_B B$. Therefore, we define $g$-factor of an electron or a hole at state $\mathbf{k}$,

$$g_k^S = \theta_k^S\left(\widehat{\mathbf{B}^{ext}}\right) \frac{\Delta E_k(\mathbf{B}^{ext})}{\mu_B B^{ext}}, \quad (8)$$

where $g_k^S$ is $g$-factor defined based on spin expectation values. $\widehat{\mathbf{B}^{\text{ext}}}$ is the unit vector along $\mathbf{B}^{\text{ext}}$. $\Delta E_k\left(\mathbf{B}^{\text{ext}}\right)$ is the energy splitting due to finite $\mathbf{B}^{\text{ext}}$. $\theta_k^S\left(\widehat{\mathbf{B}^{\text{ext}}}\right)$ is the sign of $S_{k,h}^{\exp}\left(\widehat{\mathbf{B}^{\text{ext}}}\right) - S_{k,l}^{\exp}\left(\widehat{\mathbf{B}^{\text{ext}}}\right)$, where $S_{k,h}^{\exp}\left(\widehat{\mathbf{B}^{\text{ext}}}\right)$ and $S_{k,l}^{\exp}\left(\widehat{\mathbf{B}^{\text{ext}}}\right)$ are the spin expectation value (exp) of the higher (h) and lower (l) energy band at $\mathbf{k}$ projected to the direction of $\widehat{\mathbf{B}^{\text{ext}}}$ respectively.

However, in previous theoretical studies of perovskites[35,61], $g$-factors were defined based on pseudo-spins related to the total magnetic momenta $J^{\text{at}}$, which are determined from the atomic-orbital models. The pseudo-spins can have opposite directions to the actual spins. Most previous experimental studies adopted the same convention for the signs of carrier $g$-factors. Therefore, to compare with $g$-factors obtained in previous theoretical and experimental studies, we introduce a correction factor $C^{S \to J}$ and define a new $g$-factor:

$$\widetilde{g}_k\left(\widehat{\mathbf{B}^{\text{ext}}}\right) = C^{S \to J} g_k^S. \tag{9}$$

$C^{S \to J} = m_S^{\text{at}} / m_J^{\text{at}}$ with $m_J^{\text{at}}$ and $m_S^{\text{at}}$ the total and spin magnetic momenta respectively, obtained from the atomic-orbital model[35]. $C^{S \to J}$ is independent from k-point, and is $\mp 1$ for electrons and holes respectively for CsPbBr$_3$.

$\widetilde{g}_k$ is different at different $\mathbf{k}$; therefore we define its statistically averaged value (depending on temperature $T$ and chemical potential $\mu_{F,c}$) as

$$\overline{\overline{g}} = \frac{\sum_k \left(-f_k'\right)\widetilde{g}_k}{\sum_k \left(-f_k'\right)}, \tag{10}$$

and its fluctuation amplitude as

$$\Delta \widetilde{g} = \sqrt{\frac{\sum_k \left(-f_k'\right)\left(\widetilde{g}_k - \overline{\overline{g}}\right)^2}{\sum_k \left(-f_k'\right)}}, \tag{11}$$

where $f_k'$ is the derivative of the Fermi-Dirac distribution function. Here for simplicity the band index of $f_k'$ is dropped considering both valence and conduction bands are two-fold degenerate.

We have further defined a more general $g$-factor as a tensor and its fluctuation amplitude in SI Sec. SV. For CsPbBr$_3$, we find different definitions predict quite similar values (differences are not greater than 10%).

## Analysis of e-ph matrix elements

For EY spin relaxation, in the simplified picture of Fermi's golden rule (FGR), $\tau_s^{-1}$ is proportional to the modulus square of the spin-flip scattering matrix element. As the e-ph scattering plays a crucial role in spin relaxation in CsPbBr$_3$, it is helpful to analyze the spin-flip e-ph matrix elements.

Note that most matrix elements are irrelevant to spin relaxation and we need to pick the "more relevant" ones, by defining a weight function related to occupation and energy conservation. Therefore we propose a $T$ and $\mu_{F,c}$ dependent effective modulus square of the spin-flip e-ph matrix element $|\widetilde{g}^{\uparrow\downarrow}|^2$ as

$$\overline{|\widetilde{g}^{\uparrow\downarrow}|^2} = \frac{\sum_{kq} w_{k,k-q} \sum_\lambda |g_{k,k-q}^{\uparrow\downarrow,q\lambda}|^2 n_{q\lambda}}{\sum_{kq} w_{k,k-q}}, \tag{12}$$

$$w_{k,k-q} = f_{k-q}(1-f_k)\delta\left(\epsilon_k - \epsilon_{k-q} - \omega_c\right), \tag{13}$$

where $g_{k,k-q}^{\uparrow\downarrow,q\lambda}$ is e-ph matrix element, related to a scattering event between two electronic states of opposite spins at $\mathbf{k}$ and $\mathbf{k} \cdot \mathbf{q}$ through phonon mode $\lambda$ at wavevector $q$; $n_{q\lambda}$ is phonon occupation; $f_k$ is Fermi-Dirac function; $\omega_c$ is the characteristic phonon energy specified below, and $w_{k,k-q}$ is the weight function. Here we drop band indices for simplicity, as CsPbBr$_3$ bands are two-fold Kramers degenerate and only two bands are relevant to electron and hole spin/carrier dynamics.

The matrix element modulus square is weighted by $n_{q\lambda}$ since $\tau_s^{-1}$ is approximately proportional to $n_{q\lambda}$ according to Eq. 5 of Ref. 17. This rules out high-frequency phonons at low $T$ which are not excited. $\omega_c$ is chosen as 4 meV at 10 K based on our analysis of phonon-mode-resolved contribution to spin relaxation. The trends of $|\widetilde{g}^{\uparrow\downarrow}|^2$ are found not sensitive to $\omega_c$ as checked. $w_{k,k-q}$ selects transitions between states separated by $\omega_c$ and around the band edge or $\mu_{F,c}$, which are "more relevant" transitions to spin relaxation.

We also define a $q$-resolved modulus square of the spin-flip e-ph matrix element $|\widetilde{g}^{\uparrow\downarrow}|^2(q)$ as

$$|\widetilde{g}^{\uparrow\downarrow}|^2(q) = N_k^{-1} \sum_{k\lambda} |g_{k,k-q}^{\uparrow\downarrow,q\lambda}|^2 n_{q\lambda}. \tag{14}$$

Note that for spin relaxation, only states around the band edges are relevant. Thus we restrict $|\epsilon_k - \epsilon_{\text{edge}}| < 180$ meV for the calculation of Eq. (14), which is about $7k_BT$ at 300 K relative to the band edge energy ($\epsilon_{\text{edge}}$).

## Analysis of the EY spin relaxation rate

According to FGR, the EY spin relaxation rate of an electronic state should be also proportional to the density of pair states allowing spin-flip scattering between them. Therefore, we propose a scattering density of states $D^S$ (which is $T$ and $\mu_{F,c}$ dependent),

$$D^S(T, \mu_{F,c}) = \frac{2N_k^{-2}\sum_{kq} w_{k,k-q}}{N_k^{-1}\sum_k f_k(1-f_k)}. \tag{15}$$

$D^S$ can be regarded as an effective density of spin-flip or spin-conserving e-ph transitions satisfying energy conservation between one state and its pairs (considering that the number of spin-flip and spin-conserving transitions are the same for Kramers degenerate bands).

When $\omega_c = 0$ (i.e. elastic scattering), we have $D^S = \int d\epsilon \left(-\frac{df}{d\epsilon}\right) D^2(\epsilon) / \int d\epsilon \left(-\frac{df}{d\epsilon}\right) D(\epsilon)$ with $D(\epsilon)$ density of electronic states (DOS). So $D^S$ can be roughly regarded as an weighted averaged DOS with weight $\left(-\frac{df}{d\epsilon}\right)D(\epsilon)$.

With $|\widetilde{g}^{\uparrow\downarrow}|^2$ and $D^S$, we have the approximate relation for spin relaxation rate,

$$\tau_s^{-1} \propto \overline{|\widetilde{g}^{\uparrow\downarrow}|^2} D^S. \tag{16}$$

We then discuss $\mu_{F,c}$ dependence of $\tau_s^{-1}$ at low $n_c$ limit. For simplicity, we only consider conduction electrons. At low $n_c$ limit, we have $\exp\left[(\epsilon - \mu_{F,c})/(k_BT)\right] \gg 1$, thus

$$f_{k-q}(1-f_k) \approx \exp\left(\frac{\mu_{F,c}}{k_BT}\right)\exp\left(\frac{-\epsilon_{k-q}}{k_BT}\right). \tag{17}$$

Therefore, according to Eqs. (12), (13) and (15), both $\overline{|\widetilde{g}^{\uparrow\downarrow}|^2}$ and $D^S$ are independent from $\mu_{F,c}$ (as $\exp\left(\frac{\mu_{F,c}}{k_BT}\right)$ is canceled out), so $\tau_s^{-1}$ is independent from $\mu_{F,c}$ and $n_c$ at low $n_c$ region, e.g. much lower than $10^{16}$ cm$^{-3}$ for CsPbBr$_3$. We can similarly define spin conserving matrix elements $\overline{|\widetilde{g}^{\uparrow\uparrow}|^2}$ and $|\widetilde{g}^{\uparrow\uparrow}|^2(q)$ by replacing $g_{k,k-q}^{\uparrow\downarrow,q\lambda}$ to $g_{k,k-q}^{\uparrow\uparrow,q\lambda}$ in Eq. (12) and (14). Then we have the approximate relation for carrier relaxation rate due to e-ph scattering, $\tau_p^{-1} \propto \overline{|\widetilde{g}^{\uparrow\uparrow}|^2} D^S$.

## The Hamiltonian for model SOC

In general, the Hamiltonian for model SOC reads

$$H_k^{\text{model}} = \vec{\Omega}_k^{\text{model}} \cdot \mathbf{s}_k, \tag{18}$$

where $\vec{\Omega}_k^{\text{model}}$ are Larmor precession vectors induced by $\mathbf{k}$-dependent $\mathbf{B}^{\text{in}}$. $\mathbf{s}_k$ is spin operator. With the total electronic Hamiltonian $H_k = H_{0,k} + H_k^{\text{model}}$, $\tau_s$ considering the effects of model SOC is obtained by solving the density-matrix master equation in Eq. (1).

For the Rashba field, $\vec{\Omega}_k^{\text{model}}$ in Eq. (18) is defined in the plane ($xy$ plane here) surrounding $\Gamma$ point,

$$\vec{\Omega}_k^{\text{model}} = \alpha^R f^{\text{cut}}(k/k_{\text{cut}}) \hat{z} \times \mathbf{k}, \tag{19}$$

where $\alpha^R$ is the Rashba SOC strength coefficient. $f^{\text{cut}}(k/k_{\text{cut}})$ is 1 at small $k$ but vanishes at large $k$. It is introduced to truncate the SOC fields at $k > k_{\text{cut}}$ smoothly in order to avoid unphysical band structures around first Brillouin zone boundaries. It is taken as

$$f^{\text{cut}}(k/k_{\text{cut}}) = \{\exp[10(k/k_{\text{cut}} - 1)] + 1\}^{-1}. \tag{20}$$

$k_{\text{cut}}$ is taken 0.12 bohr$^{-1}$ for CsPbBr$_3$. This value is about half of the length of the shortest reciprocal lattice vector, about 0.28 bohr$^{-1}$ for orthorhombic CsPbBr$_3$. We can see that $f^{\text{cut}}$ is almost 1 at $\mathbf{k} = \Gamma$ but almost vanishes at first Brillouin zone boundaries.

Persistent Spin Helix (PSH) was first proposed by Bernevig et al.[45] PSH has SU(2) symmetry which is robust against spin-conserving scattering. In general, for PSH SOC,

$$\vec{\Omega}_k^{\text{model}} \propto k_i \hat{j}, \tag{21}$$

where directions $i$ and $j$ are orthogonal. PSH fields are all along the same axis ($y$ axis here). We take

$$\vec{\Omega}_k^{\text{model}} = \alpha^{\text{PSH}} f^{\text{cut}}(k/k_{\text{cut}}) k_z \hat{y}, \tag{22}$$

where $\alpha^{\text{PSH}}$ is the PSH SOC strength coefficient.

## $T_{2\text{loc}}^*$ due to nuclear spin fluctuation

The Hamiltonian of hyperfine coupling between an electron spin and nuclear spins approximately reads[9,49]

$$H^{\text{hf}} = \vec{\Omega}_N \cdot \mathbf{s}, \tag{23}$$

$$\vec{\Omega}_N = V_u \sum_j A_j |\psi(\mathbf{R}_j)|^2 \mathbf{I}_j, \tag{24}$$

$$A_j = \frac{16\pi\mu_B\mu_j |u_c(\mathbf{R}_j)|^2}{3I_j}, \tag{25}$$

where $\vec{\Omega}_N$ is Larmor precession vector, related to the effective hyperfine field (called Overhauser field) generated by all nuclei and acting on electron spin. $\mathbf{s}$ is the spin operator of the electron. Eq. (24) specifically refers to the hyperfine Fermi contact interaction between an electron and nuclear spins. The sum in Eq. (24) goes over all nuclei. $\mathbf{I}_j$ is the spin operator of nucleus $j$. $V_u$ is the unit cell volume. $A_j$ is the hyperfine coupling constant considering only the Fermi contact contribution, which was assumed to be the dominant contribution in Refs. 9,49,50 for CsPbBr$_3$ and GaAs. $\mu_j$ and $I_j$ are the magnetic moment and spin of nucleus $j$, respectively. $\mu_B$ is the Bohr magneton. $\psi(\mathbf{R}_j)$ and $u_c(\mathbf{R}_j)$ are the electron envelope wave function and the electron Bloch function at the $j$-th nucleus respectively, whose product gives the electronic

wavefunction $\Phi(\mathbf{R}_j) = V_u \psi(\mathbf{R}_j) \cdot u_c(\mathbf{R}_j)$ as in Ref. 49. The normalization conditions are $\int_V |\psi(\mathbf{R}_j)|^2 dv = 1$ and

$$\int_{V_u} |u_c(\mathbf{R}_j)|^2 dv = 1. \tag{26}$$

With this definition, $|u_c(\mathbf{R}_j)|^2 \propto 1/V_u$, therefore, from Eq. (25),

$$A_j \propto 1/V_u. \tag{27}$$

The value of $A_j$ depends on the isotope of the nucleus. For CsPbBr$_3$, it was found that the relevant isotopes are $^{207}$Pb with natural abundance of about 22% for holes, and $^{79}$Br and $^{81}$Br for electrons[9]. Since the total abundance of $^{79}$Br and $^{81}$Br is almost 100% and their nuclear spins are both 3/2, $^{79}$Br and $^{81}$Br can be treated together.

According to the proportional relation in Eq. (27), $A_j$ of orthorhombic CsPbBr$_3$ is approximately 1/4 of $A_j$ of cubic CsPbBr$_3$, considering that their Bloch functions at the band edges are similar[62] (e.g., their hole Bloch functions both have significant Pb-$s$-orbital contribution), and $V_u$ of orthorhombic CsPbBr$_3$ is about 4 times of that of cubic CsPbBr$_3$. Therefore, using estimated $A_j$ of cubic CsPbBr$_3$ in Ref. 9, we obtain that $A_j$ of $^{207}$Pb for holes is about 25 $\mu$eV and $A_j$ of Br for electrons is about 1.75 $\mu$eV.

When nuclear spins are not polarized (due to $\mathbf{B}^{\text{ext}}=0$), the nuclear field is zero on average. However, due to the finite number of nuclei interacting with the localized electron, there are stochastic nuclear spin fluctuations, which are characterized by the probability distribution function[48]

$$P(\vec{\Omega}_N) = \frac{1}{(\sqrt{\pi}\sigma_{\Omega_N})^3} \exp\left(-\frac{\Omega_N^2}{\sigma_{\Omega_N}^2}\right), \tag{28}$$

where $\sigma_{\Omega_N}$ determines the dispersion of hyperfine field, and the angular brackets denotes the statistical averaging: $\langle \Omega_N^2 \rangle = 3\sigma_{\Omega_N}^2/2$. For the independent and randomly oriented nuclear spins, we have (at $\mathbf{B}^{\text{ext}}=0$)

$$\sigma_{\Omega_N}^2 = \frac{2V_u^2}{3} \sum_{j_u\xi} \alpha_\xi I_{j_u\xi}(I_{j_u\xi}+1) A_{j_u\xi}^2 \sum_c |\psi(\mathbf{R}_{j_uc})|^4, \tag{29}$$

where $j_u$ is nucleus index in the unit cell, $\xi$ is the isotope, and $c$ is the unit cell index in the whole system. $\alpha_\xi$ is the abundance of isotope $\xi$. Since $\psi(\mathbf{R}_{j_uc})^4$ usually varies slowly on the scale of a unit cell, $V_u \sum_c |\psi(\mathbf{R}_{j_uc})|^4$ can be replaced by an integral in the whole system - $\int |\psi(\mathbf{r})|^4 d\mathbf{r}$. Define $V^{\text{loc}} = 1/\int |\psi(\mathbf{r})|^4 d\mathbf{r}$. $V^{\text{loc}}$ is the localization volume. Therefore (at $\mathbf{B}^{\text{ext}}=0$),

$$\sigma_{\Omega_N}^2 = C^{\text{loc}}/V^{\text{loc}}, \tag{30}$$

$$C^{\text{loc}} = \frac{2V_u}{3} \sum_{j_u\xi} \alpha_\xi I_{j_u\xi}(I_{j_u\xi}+1) A_{j_u\xi}^2. \tag{31}$$

With $\sigma_{\Omega_N}$, $T_{2,\text{loc}}^*$ is often estimated based on FID mechanism[48–50] (Eq. (4))$T_{2,\text{loc}}^* \sim \sigma_{\Omega_N}^{-1}$.

As $\alpha_\xi$, $I_{j_u\xi}$ and $V_u$ can be easily obtained and with $A_{j_u\xi}$ estimated above, we obtain $C^{\text{loc}} \sim 180$ and $\sim 530$ nm$^3$ ns$^{-2}$ for electrons and holes respectively. $V^{\text{loc}}$ can be estimated from the localization radii $r^{\text{loc}}$ of localized carriers,

$$V^{\text{loc}} = \frac{4\pi}{3}(r^{\text{loc}})^3. \tag{32}$$

In Table S2, we listed values of the parameters used to calculate $T^*_{2,loc}$.

## Experimental synthesis

Growth of $CsPbBr_3$ single crystal: Small $CsPbBr_3$ seeds were first prepared with fresh supersaturated precursor solution at 85 °C. Small and transparent seeds were then picked and put on the bottom of the vials for large crystal growth. The temperature of the vials was set at 80 °C initially with an increasing rate of 1 °C/ h, and was eventually maintained at 85 °C. Vials were covered with glass slides to avoid fast evaporation of the DMSO. So the growth driving force is supersaturation achieved by slow evaporation of DMSO solvent. After 120–170 h, a centimeter-sized single crystal was picked from the solution, followed by wiping the residue solution on the surface.

## Experimental spin lifetime measurement

For measuring the spin lifetime in $CsPbBr_3$ single crystals, we have used the ultrafast circularly-polarized photoinduced reflectivity (PPR) method at liquid He temperature under the influence of a magnetic field. The experimental setup was described elsewhere[10,63]. It is a derivative of the well-known 'pump-probe' technique, where the polarization of the pump beam is modulated by a photoelastic modulator between left ($\delta^+$) and right ($\delta^-$) circular polarization, namely LCP and RCP, respectively. Whereas the probe beam is circularly polarized (either LCP or RCP) by a quarter-wave plate. The transient change in the probe reflection, namely c-PPR(t), was recorded. The 405 nm pump beam, having 150 femtoseconds pulse duration at 80 MHZ repetition rate, was generated by frequency doubling the fundamental at 810 nm from the Ti:Sapphire laser (Spectra Physics) using a SHG BBO crystal. The 533 nm probe beam was generated by combining the 810 nm fundamental beam with the 1560 nm infrared beam from an OPA (optical parametric amplifier) onto a BBO type 2 SFG (Sum Frequency Generation) crystal. The pump/probe beams having average intensity of 12 Wcm$^{-2}$ and 3 Wcm$^{-2}$, respectively were aligned onto the $CsPbBr_3$ crystal that was placed inside a cryostat with a built-in electro-magnet that delivered a field strength, B up to 700 mT at temperatures down to 4 K. Using this technique we measured both t-PPR responses at both zero and finite B to extract the B-dependent electron and hole spin lifetimes. From the c-PPR(B,t) dynamics measured on (001) facet with B directed along [010][63] (see example c-PPR(B,t) dynamics in SI Fig. S15), we could obtain the electron and hole $T^*_2$ by fitting the transient quantum beating response with two damped oscillation functions:

$$A_1 e^{\frac{-t}{T^*_{2,e}}}cos(2\pi f_1 t + \phi_1) + A_2 e^{\frac{-t}{T^*_{2,h}}}cos(2\pi f_2 t + \phi_2), \qquad (33)$$

where $T^*_{2,e}$ and $T^*_{2,h}$ are the spin dephasing times of the electrons and holes; f1 and f2 are the two QB frequencies that can be obtained directly from the fast Fourier transform of the c-PPR dynamics.

## Data availability

The input files of all simulations (including the ground-state DFT simulations, Wannier fitting and interpolation, and the real-time density-matrix simulations), python post-processing scripts, example output files and necessary source data files (for plotting) generated for this study are available in the SI repository (https://github.com/Ping-Group-UCSC/Data-NC-spin-2023).

## Code availability

The codes are available through open-source software, JDFTx[56] and QUANTUM ESPRESSO[65], or from authors upon request.

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

## Acknowledgements

We acknowledge support for the theoretical development of spin dynamics in the presence of large spin-orbit coupling and magnetic field by the computational chemical science program within the Office of Science at DOE under grant No. DE-SC0023301. The experimental measurements of spin dynamics are supported as part of the Center for Hybrid Organic-Inorganic Semiconductors for Energy (CHOISE), an Energy Frontier Research Center funded by the Office of Basic Energy Sciences, Office of Science within the US Department of Energy (DOE). This research used resources of the Center for Functional Nanomaterials, which is a US DOE Office of Science Facility, and the Scientific Data and Computing center, a component of the Computational Science Initiative, at Brookhaven National Laboratory under Contract No. DE-SC0012704, the lux supercomputer at UC Santa Cruz, funded by NSF MRI grant AST 1828315, the National Energy Research Scientific Computing Center (NERSC) a U.S. Department of Energy Office of Science User Facility operated under Contract No. DE-AC02-05CH11231, and the Extreme Science and Engineering Discovery Environment (XSEDE) which is supported by National Science Foundation Grant No. ACI-1548562[64].

## Author contributions

J.X., K.L., and M.F. performed the ab initio calculations; J.X., K.L., R.S., and Y.P. analyzed the theoretical results. J.X. and R.S. implemented the computational codes. U.N.H., J.H., and V.V. did the experimental measurements. Y.P. designed and supervised all aspects of the study. J.X and Y.P. wrote the first draft of the manuscript. All authors contributed to the writing of the manuscript.

## Competing interests

The authors declare no competing interests.
