## [Peer Review File · Nature Communications]

Reviewers' Comments:

Reviewer #1:

Remarks to the Author:

In the paper "How Spin Relaxes in Bulk Halide Perovskites" by J. Xu et al. the authors calculate for the bulk all-inorganic lead halide perovskite CsPbBr₃ the spin lifetime and spin dephasing time. The spin lifetime dependence on temperature and carrier density, in respect of the carrier-phonon, the carrier-carrier and the carrier-impurity interactions is highlighted, and for the spin dephasing time a k-dependent g-factor and its spread is calculated, in regard of the same interactions. The paper is based on an ab-initio approach, backed by single crystal CsPbBr₃ pump probe data and some collected literature values. The results are very interesting and predictive for future perovskite development. However, a few questions arise:

- 1) First of all, for me it is difficult to gain a complete overview of the studied situation. Basically the low and high temperature regimes are discussed and a high emphasis placed on the low temperature regime, however only in Fig. 3 a logarithmic x-scale chosen which allows to visualize them both simultaneously. Or magnifications as used in Fig. 2 or Fig. S9 would be good. Further, the authors have chosen in both cases of the spin lifetime and spin dephasing time, their initial model to present the experimental data. However, this initial model than they correct by carrier impurity interactions. For a better interpretation of the completeness of the presented results a comparison with a combined model would be highly requested. In particular, the evaluated amount of $1 \times 10^{18} \text{ cm}^{-3}$ impurities, for their sample, leading to a saturating spin lifetime at small temperatures, would give a strongly suppressed spin dephasing according to Fig. 6d. Which makes the evaluation inconsistent.
- 2) Staying with the experimental data, the model of g-factor Fig. 5 predicts a low hole g-factor of +0.25 for the carrier concentration of $1 \times 10^{18} \text{ cm}^{-3}$ and a high dependence on the carrier concentration in this carrier concentration regime. I couldn't find the obtained g-factors in Fig. S10, to check this finding.
- 3) Overall the theoretical model was benchmarked with TMDC data in SIV (and also the spin lifetime temperature dependence (main text) was compared with TMDC properties), which makes me wondered why it wasn't compared with other perovskites structures or the well known III-V or II-VI semiconductors? This could be explained in section SIV. Further a comparison of the observed spin dynamics with the well-studied ones like GaAs could be helpful (e.g. see Phys. Rev. B 66, 245204)
- 4) The carrier concentration is assumed to lay on a level of $1 \times 10^{18} \text{ cm}^{-3}$ in the experiment based on the excitation power. As n_c is an important parameter for the results, could it be tuned to verify the findings? For instance, the g-factor and T1 temperature dependence should than change dramatically with the excitation power. From the signal to noise ratio, in Fig. S10, a lower excitation seems to be feasible.
- 5) The authors also include findings on CsPbBr₃ nanocrystals (NC), without any further comments. Though NCs are not in focus of the paper, if to include them as evidence for the model, one should comment how confinement would influence the model.
- 6) The calculations based on the quantum espresso code are a black box. For reproduction, a clean mathematical description and parameters are needed and for understanding, a more descriptive analysis should be provided.
- 7) The paper is self consistent with the Ansatz to only consider EY as spin relaxation mechanism with the conclusion that this dominates, but not complete in term of other mechanisms, like most importantly Dykanov-Perel. For a better evaluation of the calculated e-ph coupling strengths, EY and DP mechanisms need to be compared. To cite Ref. [S25] "However, in the case of the EY mechanism, the spin coherence time will be long if the electron-phonon coupling is small, whereas for the DP mechanism, the faster the momentum relaxation, the slower the spin dephasing."
- 8) At several points the influence of the nuclear spin bath is neglected in the main text, though it is known from literature to be of an huge importance for the experimentally observed spin dynamics. "It may originate from nuclear spin fluctuation, magnetic impurities, carrier localization, chemical potential fluctuation, etc.[10, 28] in samples, which are however beyond the scope of this work.". One is bulk CsPbBr₃ if this is out of scope, what is the scope of the work?
- 9) I wonder how the presented results, which were mainly discussed for CsPbBr₃, can be generalized for the general class of Bulk Halide Perovskite, mentioned in the title. For instance,

how the results can be transferred with the exchange of A or B side cation, for instance to change to tin based halide perovskites or the photovoltaic archetype material hybrid organic-inorganic lead halides like methylammonium MAPbI₃ perovskites. In particular the exchange of lead to tin, should lead to a drastic change of the spin orbit constant, thus have a high impact on the spin dynamics. For the hybrid organic to all inorganic comparison, for instance the work could be compared with the paper "Unravelling the Spin Relaxation Mechanism in Hybrid Organic–Inorganic Perovskites" J. Phys. Chem. C 2019, 123, 14701–14706.

10) As minor remark, an (empirical) master equation which includes all terms, like $1/\tau_s = 1/t(e-e) + 1/t(e-ph) + 1/T(e-i)$ would be of high usage. Or a 2D plot T₁ upon T and n_c in one could be useful.

To conclude, the examination of spin relaxation times for CsPbBr₃ is a very interesting topic. However, I worry that the claim to address all bulk halide perovskite structures is not yet fulfilled, the general claim of a full presentation of spin relaxation lacks completeness in respect of other mechanisms and further the experimental verification of the results not clearly presented.

Reviewer #2:

Remarks to the Author:

This manuscript reports a theoretical investigation on the spin relaxation and dephasing mechanism of carriers in halide perovskite via FPDM approach and probed the dependence of τ_s on different T and n_c, external fields, carrier density, and impurities. The results reveal that $\tau_s \sim 1$ and $\tau_p \sim 1$ (due to e-ph scattering) are proportional to the modulus square of spin-flip and spin-conserving matrix elements (ME), respectively. More importantly, the correlation between Landé g-factors, B-induced energy splitting $\Delta E_k(B)$, and Larmor precession frequency, is elaborated, which is not present in previous studies. The spin relaxation in pristine CsPbBr₃ at low T is predicted to be ultraslow and magnetic-field effects are only significant under T < 20 K. I recommend its publication before the following issues are addressed:

1. I have some concerns about the methodology. The author proposes that the FPDM method is used in the article, but discussion about the applicability of different mechanisms, such as EY and DP mechanisms in CsPbBr₃ should be enhanced.
2. I want to know whether it is feasible to build a clearer structure-function relationship, i.e., between the atomic structure and spin lifetimes, to resolve the contributions of cation, Pb, and Br atoms.
3. On P. 3, the authors state that at low T and low n_c, CsPbBr₃ has a relatively long spin lifetime. Similar to TMDs, this phenomenon is attributed to the spin-valley locking, where the strength of the spin-valley locking has been quantitatively analyzed by measuring the spin lifetimes intervalley and intravalley. How about this in CsPbBr₃?
4. The authors claim "compute the spin relaxation time (T₁) and ensemble spin dephasing time (T_{2*})" in the Abstract, however, the manuscript does not give a clear discussion of spin dephasing time or data to support the statement of "ensemble spin dephasing time (T_{2*})" in Abstract.
5. I did not find the difference between panels (c) and (e), (d), and (f) in FIG S3. They are the same, please check.
6. The order of pictures described in the manuscript is chaotic, the in-text citation of Fig.1 (b) precedes Fig.1 (a).
7. On page 3, how to explain "This is contradictory to the simple assumption frequently employed in previous experimental studies."
8. On page 4, font error in the first paragraph.

Below, we repeat the reviewer's comments in black italic and present our responses point-by-point in blue color.

REVIEWER COMMENTS

Reviewer #1 (Remarks to the Author):

"In the paper "How Spin Relaxes in Bulk Halide Perovskites" by J. Xu et al. the authors calculate for the bulk all-inorganic lead halide perovskite CsPbBr₃ the spin lifetime and spin dephasing time. The spin lifetime dependence on temperature and carrier density, in respect of the carrier-phonon, the carrier-carrier and the carrier-impurity interactions is highlighted, and for the spin dephasing time a k-dependent g-factor and its spread is calculated, in regard of the same interactions. The paper is based on an ab-initio approach, backed by single crystal CsPbBr₃ pump probe data and some collected literature values. The results are very interesting and predictive for future perovskite development. However, a few questions arise:"

We thank the reviewer's appreciation of our work and the comment that 'our results are very interesting and predictive for future perovskite development'.

Moreover, with the reviewer's constructive feedback and helpful suggestions, we have greatly improved the generality of our manuscript and further clarified several points in our original manuscript.

Importantly, two major revisions were made:

(i) We added a subsection "Inversion symmetry broken (ISB), composition effects and hyperfine coupling" and a related figure -- Fig. 6 on Page 8-9 of the revised manuscript. The FPDM simulations and related discussions for the ISB and chemical composition effects improve the generality of our work. The theoretical estimates and related discussions of hyperfine coupling effects provide a more complete description of spin relaxation/dephasing in halide perovskite.

(ii) We added a brief introduction of T_1 and T_2^* and briefly discussed the possible physical mechanism that limits T_1 and T_2^* of bulk carriers in halide perovskites.

These were added just above the subsection “Spin lifetimes at zero magnetic field” on the right column of page 2. These discussions significantly clarify our theoretical calculations and discussions.

“1) First of all, for me it is difficult to gain a complete overview of the studied situation. Basically the low and high temperature regimes are discussed and a high emphasize placed on the low temperature regime, however only in Fig. 3 a logarithmic x-scale chosen which allows to visualize them both simultaneously. Or magnifications as used in Fig. 2 or Fig. Fig. S9 would be good.”

We thank the reviewer for this constructive suggestion.

We added a figure using log-scale for the x-axis to highlight the low-T region of τ_s (without impurities) versus T. See SI Fig. S6. Moreover, the new Fig. 1c in the revised manuscript shows τ_s with and without impurities compared with the experiments in the low T region.

“Further, the authors have chosen in both cases of the spin lifetime and spin dephasing time, their initial model to present the experimental data. However, this initial model than they correct by carrier impurity interactions. For a better interpretation of the completeness of the presented results a comparison with a combined model would be highly requested. In particular, the evaluated amount of $1 \times 10^{18} \text{ cm}^{-3}$ impurities, for their sample, leading to a saturating spin lifetime at small temperatures, would give a strongly suppressed spin dephasing according to Fig. 6d. Which makes the evaluation inconsistent.”

We added experimental data to compare with our theoretical T_1 and $1/T_2^*$ calculations accounting for impurities in Fig. 1c and Fig. 5d, respectively in the revised manuscript (which correspond to Fig. 2a and Fig. 6d in the original manuscript, since we merged the original Fig. 1 and Fig. 2).

Through the comparisons, we found that impurity scattering at high impurity density n_i (e.g. 10^{18} cm^{-3}) reduces the theoretical τ_s to a few ns below 10 K, matching our measured T_2^* . However, the magnetic-field dependence of the theoretical T_2^* with

relatively high n_i (e.g. 10^{17} cm^{-3}) disagrees with that of our measured T_2^* at 4 K. Therefore, including relatively strong impurity scattering in FPDM simulations cannot explain the two sets of our measured T_2^* .

Therefore, we made the following revisions:

(i) We commented on the discrepancy between our theoretical τ_s without impurities and our measure T_2^* below 10 K using the following sentences on the right column of page 3 in the subsection “Spin lifetimes at zero magnetic field”:

“The discrepancy is possibly due to nuclear-spin-induced spin dephasing of carriers Therefore, the discrepancy between our theoretical τ_s and our measured T_2^* below 10 K is probably not explained by the impurity scattering effects.”

(ii) We added the following sentence at the end of the subsection “Landé g-factor and transverse-magnetic-field effects” (on the right column of page 8): “From Fig. 5d, we find that with relatively strong impurity scattering (e.g, with $10^{17} \text{ cm}^{-3} V_{\text{ph}}$ neutral impurities), the B^{ext} -dependence of τ_s is in disagreement with experiments, indicating that impurity scattering is probably weaker in those experiments.”

“2) Staying with the experimental data, the model of g-factor Fig. 5 predicts a low hole g-factor of +0.25 for the carrier concentration of $1 \times 10^{18} \text{ cm}^{-3}$ and a high dependence on the carrier concentration in this carrier concentration regime. I couldn't find the obtained g-factors in Fig. S10, to check this finding.”

Firstly, we want to clarify that we did not intent to compare with experimental g factor value quantitatively because of the sensitivity on DFT exchange correlation functional (Vxc) or electronic structure methods as we explained in SI section SV. Therefore we didn't present experimental g factor of this system specifically, but we have cited prior g factor values of this system.

On the other hand, the carrier density dependence of g factor and the g factor fluctuations are less sensitive to the choice of Vxc or electronic structure methods. In particular, the g factor fluctuation (Δg) determines the spin dephasing rate ($T_2^{*(-)}$) slope as a function of external B field in the relatively large B field range, which we have compared with experiments in main text Figure 5.

Second, our hole averaged g factor at PBE level is -0.25 if including the sign, at carrier density of $1 \times 10^{18} \text{ cm}^{-3}$. We have clarified this point by replotting averaged g factor with the sign in Figure 4b of the revised manuscript. This value is likely underestimated in comparison with past experimental measurements. As shown in Figure S5 and related discussions in the paragraph, the hole g factor is the closest to experimental results with EV93PW91 functional.

“3) Overall the theoretical model was benchmarked with TMDC data in SIV (and also the spin lifetime temperature dependence (main text) was compared with TMDC properties), which makes me wondered why it wasn't compared with other perovskites structures or the well known III-V or II-VI semiconductors?”

This could be explained in section SIV. Further a comparison of the observed spin dynamics with the well-studied ones like GaAs could be helpful (e.g. see Phys. Rev. B 66, 245204)”

Note that we have benchmarked our spin lifetime calculations against several systems in our previous work (Refs. 18, 19 and 21 in the revised manuscript) including Si, Fe, TMDs, graphene-hBN, GaAs. Such benchmarks were very thoroughly discussed including temperature, carrier density, external field dependence, and they compared well with experimental studies.

We have reemphasized in our introduction that “FPDM approach was applied to disparate materials including silicon, iron, transition metal dichalcogenides (TMDs), graphene-hBN, GaAs, in good agreement with experiments [18, 19, 21].” to the last paragraph of the introduction on the left column of page 2.

The only thing not included in our previous work is the g-factor calculation. This is what we benchmarked against TMDC in this work.

The main reason we compared CsPbBr_3 data with TMDs data is that spin relaxation in TMDs is dominated by EY mechanism (or spin-flip scattering), the same as in bulk CsPbBr_3 . Moreover, carrier-density dependence of spin lifetime of TMDs was studied both theoretically by us and experimentally. Therefore, these studies are good references for understanding spin relaxation and its carrier-density dependence in bulk CsPbBr_3 . In contrast, spin relaxation in GaAs is dominated by other mechanisms such as DP mechanism, but not EY.

We have changed the sentence “Such phenomenon was reported previously for monolayer WSe₂[19,27].” to “Such phenomenon was reported previously for monolayer WSe₂[19,31], where spin relaxation is dominated by EY mechanism, same as in CsPbBr₃.” on the left column of page 3. This change emphasizes the reason why we compare CsPbBr₃ with TMDs.

“4) The carrier concentration is assumed to lay on a level of $1 \times 10^{18} \text{cm}^{-3}$ in the experiment based on the excitation power. As n_c is an important parameter for the results, could it be tuned to verify the findings? For instance, the g -factor and T_1 temperature dependence should than change dramatically with the excitation power. From the signal to noise ratio, in Fig. S10, a lower excitation seems to be feasible.”

We thank the reviewer for this suggestion; we have indeed tried that. However experimentally it is very difficult to observe a small g -value change with the excitation intensity, I since $n_c(I)$ depends as I^p with $p < 0.3$ due to the bimolecular (or higher) recombination process of the photocarriers. Note that the steady state photocarrier density n_c (i.e. background density) is much smaller than the peak density $n_c(t=0)$ generated at $t=0$, due to fast photocarrier density decay (~ 1 ns). It is much better to change the carrier density by doping. When we did that, we found indeed that the electron g -value decreases and the spin lifetime decreases dramatically.

Additionally, we found other examples where spin lifetime decreases while increasing carrier density. spin lifetime of monolayer WSe₂ decreases with increasing n_c was reported in both previous experimental work (Fig. 2e of Phys. Rev. Mater. 5, 044001 (2021)) and our previous theoretical study (SI Fig. S12 of Phys. Rev. B 104, 184418 (2021)), we note that T_1 of halide perovskites decrease with increasing pump power or fluence were reported in several previous experiment works, including Nano Lett. 2015, 15, 1553–1558; J. Am. Chem. Soc. 2021, 143, 46, 19438–19445; Nano Lett. 2023, 23, 205–212; Nat. Commun. 11, 5665 (2020).

We added a related sentence “The trend of T_1 decrease with n_c is consistent with the experimental observation of T_1 decrease with the pump power/fluence in halide perovskites[27–30].” in the subsection “Spin lifetimes at zero magnetic field” on the left column of page 3.

“5) The authors also include findings on CsPbBr₃ nanocrystals (NC), without any further comments. Though NCs are not in focus of the paper, if to include them as evidence for the model, one should comment how confinement would influence the model.”

We thank the reviewer for this suggestion. We added a related comment in the caption of Fig. 1: “In Ref. 30, it was declared that quantum confinement effects do not modify the spin relaxation/dephasing significantly (see Table 1 there); therefore their T_1 data can be compared with our theoretical results.”

“6) The calculations based on the quantum espresso code are a black box. For reproduction, a clean mathematical description and parameters are needed and for understanding, a more descriptive analysis should be provided.”

We apologize that our description was not clear to the reviewer. We clarify it as follows.

Firstly, we note that the calculations we performed in this work are nontrivial; we developed our own in-house codes based on density matrix dynamics with quantum scattering from first principles, as explained in detail through model and numerical implementation in our previous studies (Phys. Rev. B 104, 184418 (2021)).

Secondly, we have described in the SI, Sec SII our computational details; most of our simulations are done using JDFTx code and the in-house DMD code (Density-Matrix Dynamics), interfaced to JDFTx.

QuantumEspresso code is only employed to provide three parameters – Born effective charges, Z , electronic dielectric constants, ϵ_∞ and macroscopic static dielectric constant, ϵ_0 as used in the screened Coulomb potential for electron-electron scatterings. Only a very small part of the calculation.

To clarify the details further in the manuscript, we therefore added a sentence “The simulations of Z , ϵ_∞ and ϵ_0 employ the commonly-used method developed in Ref. 13 based on DFPT and use the implementations in QuantumESPRESSO.” in SI Sec. SII Computational details and at the end of page 2 in SI.

“7) The paper is self consistent with the Ansatz to only consider EY as spin relaxation mechanism with the conclusion that this dominates, but not complete in term of other mechanisms, like most importantly Dykanov-Perel. For a better evaluation of the calculated e-ph coupling strengths, EY and DP mechanisms need to be compared. To cite Ref. [S25] “However, in the case of the EY mechanism, the spin coherence time will be long if the electron–phonon coupling is small, whereas for the DP mechanism, the faster the momentum relaxation, the slower the spin dephasing.”

We agree that the DP mechanism is very important for spin relaxation in halide perovskites.

However, we need to clarify that we have not made the ANSATZ that only EY is considered for bulk carriers. For bulk CsPbBr₃, since its most stable structure phase is inversion symmetric, then the DP mechanism can not be dominant for the spin relaxation of electron or hole carriers in bulk CsPbBr₃, unless inversion-symmetry-broken happens. Note that whenever inversion symmetry is present, at zero magnetic field $B^{\text{ext}}=0$, all bands are Kramers degenerate. Thus, there is no Larmor spin precession, which is necessary for the DP mechanism.

Furthermore, at nonzero external B field (B^{ext}), Kramers degeneracy is broken and spin can precess. As we discussed in subsection “Landé g-factor and transverse magnetic fields effects”, DP spin decay can present at a weak B^{ext} which ensures $\tau_p \Delta \Omega \ll 1$ (strong scattering limit) is satisfied. While free induction decay (FID) spin decay can present at large B^{ext} .

To have DP mechanism at $B^{\text{ext}}=0$, the inversion symmetry needs to be broken so that there can be non-zero internal magnetic field causing spin precession. Considering that inversion symmetry broken (ISB) may happen in halide perovskites due to ferroelectric polarization, strain, surface, applying electric fields, etc. we studied the ISB effects through *ab initio* simulations using our FPDM approach. Since One of the most important effects of ISB is inducing k-dependent SOC fields, to understand ISB effects, we simulate τ_s with two important types of SOC fields - Rashba and PSH (persistent spin helix). These SOC terms were introduced to the electronic Hamiltonian perturbatively.

Through our FPDM simulations, we found that Rashba SOC induces an additional spin relaxation channel (can be DP or FID depending on $\tau_p \Delta \Omega$) in addition to the EY

channel and likely reduces τ . While PSH SOC can enhance τ , but the EY mechanism still dominates via the PSH SOC.

Details of our FPDM simulations and related discussions for the ISB effects are discussed in the new subsection “Inversion symmetry broken (ISB), composition effects and hyperfine coupling” and the new figure Fig. 6 in the revised manuscript on Page 8-9, as well in the technical subsection “The Hamiltonian for model SOC” in the “Methods” part. .

“8) At several points the influence of the nuclear spin bath is neglected in the main text, though it is known from literature to be of an huge importance for the experimentally observed spin dynamics. "It may originate from nuclear spin fluctuation, magnetic impurities, carrier localization, chemical potential fluctuation, etc.[10, 28] in samples,

which are however beyond the scope of this work." One is bulk CsPbBr3 if this is out of scope, what is the scope of the work?”

We previously focused on spin relaxation/dephasing of bulk (or itinerant or delocalized) carriers, for which hyperfine coupling between carrier spin and nuclear spin is usually considered unimportant, since the averaged nuclear-spin-induced magnetic field felt by bulk carriers is tiny when the nuclear spins are weakly polarized.

We agree that hyperfine coupling is important to spin dynamics at low T. We added two paragraphs discussing the effects of hyperfine coupling on spin decay of localized and bulk carriers in halide perovskites. We estimated T_2^* of localized carriers $T_{2,loc}^*$ due to polarons, ionized impurities, etc.. T_2^* is estimated based on the model relation commonly used in literature (e.g., Phys. Rev. B 102, 235413 (2020), Phys. Rev. B 84, 085304 (2011), Phys. Rev. B 65, 205309 (2002), Nat. Commun. 10, 1 (2019)). We obtained $T_{2,loc}^* \sim 0.35-4.6$ ns and $\sim 0.6-8.0$ ns for electrons and holes respectively, consistent with the experimental range of T_2^* from a few ps to tens of ns for halide perovskites.

We also mentioned an additional possible spin dephasing channel for the bulk carriers, namely the fluctuation of the hyperfine field for bulk carriers at different k-

points may lead to spin dephasing when the nuclear spins are polarized along a transverse B^{ext} .

The added sentences are: “Above we focus on spin relaxation/dephasing of bulk (or itinerant or delocalized) electrons, for which hyperfine coupling is usually unimportant[6, 50]. This effect is however rather complicated (probably requiring the difficult L contribution[58] to the hyperfine coupling) beyond the scope of this work.” These can be found in the last two paragraphs of the subsection “Inversion symmetry broken (ISB), composition effects and hyperfine coupling” on page 9.

“9) I wonder how the presented results, which were mainly discussed for CsPbBr₃, can be generalized for the general class of Bulk Halide Perovskite, mentioned in the title. For instance, how the results can be transferred with the exchange of A or B side cation, for instance to change to tin based halide perovskites or the photovoltaic archetype material hybrid organic-inorganic lead halides like methylammonium MAPbI₃ perovskites. In particular the exchange of lead to tin, should lead to a drastic change of the spin orbit constant, thus have a high impact on the spin dynamics. For the hybrid organic to all inorganic comparison, for instance the work could be compared with the paper "Unravelling the Spin Relaxation Mechanism in Hybrid Organic–Inorganic Perovskites" J. Phys. Chem. C 2019, 123, 14701–14706.”

We thank the reviewer for these suggestions. Note that similar experimental results from our laboratory were already published for MAPI (ref. 11); we are currently finishing theoretical part of the work. We are measuring other 3D HOIPs at the present time and hope that the theoretical calculation would fit the experiment as well.

We added our FPDM theoretical results of T_1 of MAPbBr₃, which is a typical organic-inorganic halide perovskite whose spin dynamics have been extensively studied. We considered the inversion-symmetric orthorhombic phase, the same as that of CsPbBr₃ in our work. Overall, we find that by replacing Cs to MA molecule, T_1 only changes by 16%-72% in a wide temperature range of 4-300 K.

The related added sentences are: “It is crucial to understand the composition effects A systematic comparison of MA/CsBX₃(B= Sn, Pb, X= Cl, Br, I) is of great importance and will be done in our future work.”

Since the spin relaxation/dephasing simulations of halide perovskites are much more computationally costly than simple materials like TMDs, as an initial work of halide perovskites, we focus on FPDM simulations of CsPbBr₃ with discussions of ISB and composition effects. We are studying spin relaxation/dephasing in both inversion-symmetric and inversion-symmetry-broken MAPbX₃. We plan to study CsPbI₃ and CsSnX₃ in the future to understand better the composition effects

“10) As minor remark, an (empirical) master equation which includes all terms, like $1/\tau_s = 1/t(e-e) + 1/t(e-ph) + 1/T(e-i)$ would be of high usage. Or a 2D plot T_1 upon T and nc in one could be useful.”

This may be ok for a system dominated by EY, not for systems dominated by DP. This is analogous to Mathessien’s rule for carriers relaxation.

Our method includes all effects from quantum master equation. In any case, we added this plot to the SI in Fig. S11 with corresponding discussions in SI Sec. SVI. “Spin relaxation times”. We indeed found that the total spin relaxation rate $1/\tau_s^{\text{tot}}$ agrees with the sum of the rates due to individual scatterings, i.e., $1/\tau_s^{\text{tot}} \approx 1/\tau_s^{\text{e-ph}} + 1/\tau_s^{\text{e-i}} + 1/\tau_s^{\text{e-et}}$.

“To conclude, the examination of spin relaxation times for CsPbBr₃ is a very interesting topic. However, I worry that the claim to address all bulk halide perovskite structures is not yet fulfilled, the general claim of a full presentation of spin relaxation lacks completeness in respect of other mechanisms and further the experimental verification of the results not clearly presented.”

We thank the critical comments and helpful suggestions of the reviewer that have helped us to improve the presentation of our calculations.

We added a new subsection “Inversion symmetry broken (ISB), composition effects and hyperfine coupling” and a new figure Fig. 6 showing theoretical results of T_1 when considering Rashba and PSH SOC. With these new contents and many other

improvements, we had generalized our work for other halide perovskites with ISB and different compositions beyond CsPbBr₃, and for other mechanisms, e.g., spin dephasing of localized electrons through spin precessions about randomly-distributed nuclear-spin (magnetic) fields induced by hyperfine coupling.

In our original manuscript we had shown that our calculated T_1 and its temperature dependence are in good agreement with experiments. Our calculated T_2^* at relatively high magnetic fields are also in agreement with the experiments. In the revised manuscript, we added new citations which dealt with T_1 in halide perovskites and found that it decreases with increasing pump power/fluence, consistent with the trend of theoretical T_1 decrease with increasing carrier density predicted by our FPDM approach. Importantly, as we discussed above, we were studying the effects of carriers density by varying the photoexcitation intensity. We found that when the photocarriers density increases, the electron g -factor decreases and also the spin lifetime decreases dramatically.

Moreover, as we clarified in the revised manuscript, our approach had been benchmarked against disparate materials including silicon, iron, TMDs, graphene-hBN, GaAs. In addition we obtained carrier density dependences in agreement with experiments for WSe₂, graphene on hBN substrate and GaAs. Therefore, our approach is reliable for predicting spin lifetimes and their carrier density dependences. See our theoretical results of WSe₂ in SI Fig. S12 in Phys. Rev. B 104, 184418 (2021) compared with Fig. 2e of Phys. Rev. Mater. 5, 044001 (2021). See theoretical results of graphene-hBN in Fig. 3a in Phys. Rev. B 104, 184418 (2021) compared with Nano Lett. 16, 3533 (2016) and Phys. Rev. B 86, 161416(R) (2012). See theoretical results of GaAs in Fig. 5a of Phys. Rev. B 104, 184418 (2021) without considering nuclear spin effects compared with Phys. Rev. Lett. 80, 4313 (1998).

Overall, with our new results, discussions and clarifications, we believe that our work provides fundamental insights on spin relaxation and dephasing of electron or hole carriers in the class of halide perovskites.

Reviewer #2 (Remarks to the Author):

“This manuscript reports a theoretical investigation on the spin relaxation and dephasing mechanism of carriers in halide perovskite via FPDM approach and probed the dependence of τ_s on different T and n_c , external fields, carrier density, and impurities. The results reveal that $\tau_s \sim 1$ and $\tau_p \sim 1$ (due to e-ph scattering) are proportional to the modulus square of spin-flip and spin-conserving matrix elements (ME), respectively. More importantly, the correlation between Landé g-factors, B-induced energy splitting $\Delta E_k(B)$, and Larmor precession frequency, is elaborated, which is not present in previous studies. The spin relaxation in pristine CsPbBr₃ at low T is predicted to be ultraslow and magnetic-field effects are only significant under $T < 20$ K. I recommend its publication before the following issues are addressed.”

We greatly appreciate the reviewer’s positive opinion on our work. We have greatly improved the generality of our manuscript and clarified many points that were somewhat unclear in our original manuscript. Importantly, two major revisions were made:

(i) A new subsection “Inversion symmetry broken (ISB), composition effects and hyperfine coupling” and a new figure -- Fig. 6 were added in the revised manuscript on Page 8-9. The FPDM simulations and related discussions for the ISB and component effects improve the generality of our work. The theoretical estimates and related discussion of hyperfine coupling effects makes our work on spin relaxation/dephasing in halide perovskite more complete.

(ii) We added brief introduction of T_1 and T_2^* and briefly discussed the possible physical mechanisms that limits T_1 and T_2^* of bulk carriers in halide perovskites placed just above the subsection “Spin lifetimes at zero magnetic field” on the right column of page 2. These clarify the theoretical results and discussions to a casual reader.

“1. I have some concerns about the methodology. The author proposes that the FPDM method is used in the article, but discussion about the applicability of different mechanisms, such as EY and DP mechanisms in CsPbBr₃ should be enhanced.”

We agree that DP and other mechanisms are in general very important for spin relaxation in halide perovskites.

We would like to clarify first that our FPDM approach was successfully applied to both EY and DP systems. To emphasize this, we added a related sentence “FPDM approach was applied to disparate materials including silicon, iron, transition metal dichalcogenides (TMDs), graphene-hBN, GaAs, good agreement with experiments were achieved.[18, 19, 21]” to the last paragraph of the introduction on the left column of page 2.

For bulk CsPbBr₃, since its most stable structure phase is inversion symmetric, DP mechanism is irrelevant in spin relaxation of excess carriers in bulk CsPbBr₃. The reason for this is that the inversion symmetry all bands are Kramers degenerate at zero magnetic field $B_{\text{ext}}=0$. Thus, there is no Larmor spin precession, which is necessary for the DP mechanism.

As we discussed in subsection “Landé g-factor and transverse magnetic fields effects”, DP spin decay may be present at a weak external magnetic field B_{ext} when $\tau_p \Delta\Omega \ll 1$ (strong scattering limit) is satisfied. While free induction decay (FID) spin decay can present at large B_{ext} .

To have DP or FID mechanism at $B_{\text{ext}}=0$, the inversion symmetry should be broken so that there can be random-like spin precession for DP/FID mechanism. Considering that inversion symmetry broken (ISB) may happen in many halide perovskites, we studied the ISB effects through ab initio simulations using our FPDM approach. Since one of the most important effects of ISB is inducing k-dependent SOC fields, to understand ISB effects, we simulate τ_s with two important types of SOC fields - Rashba and PSH (persistent spin helix), which are introduced by an additional SOC term to the electronic Hamiltonian perturbatively.

Through our FPDM simulations, we found that Rashba SOC induces an additional spin relaxation channel (can be DP or FID depending on $\tau_p \Delta\Omega$) in addition to the EY and this probably reduced τ_s . While PSH SOC may enhance τ_s if its strength is large enough but the EY mechanism still dominates with PSH SOC.

Details of our FPDM simulations and related discussions for ISB effects are given in the new subsection “Inversion symmetry broken (ISB), composition effects and hyperfine coupling” and the new figure Fig. 6 in the revised manuscript on Page 8-

9. A technical subsection “The Hamiltonian for model SOC” was also added in “Methods” part to give more technical details with formulae.

“2. I want to whether it is feasible to build a clearer structure-function relationship, i.e., between the atomic structure and spin lifetimes, to resolve the contributions of cation, Pb, and Br atoms.”

Since spin lifetime is a very complicated quantity, it is difficult to build a clear relation between atomic structure and spin lifetime.

We still made efforts in several aspects:

(i) We studied how inversion symmetry broken (ISB) can affect spin lifetimes as we mentioned above.

(ii) We added our FPDM theoretical results of T_1 of MAPbBr_3 , which is a typical organic-inorganic halide perovskite whose spin dynamics had been extensively studied. We considered the inversion-symmetric orthorhombic phase, the same as CsPbBr_3 in our work. Overall, we find that by replacing Cs with MA molecule, T_1 only changes by 16%-72% in a wide temperature range 4-300 K.

The related sentences are: “It is crucial to understand the composition effects A systematic comparison of MA/CsBX_3 ($\text{B} = \text{Sn, Pb}$, $\text{X} = \text{Cl, Br, I}$) is of great importance and will be our future work.”

(iii) We added atomic-orbital-projected electronic density of states (DOS) and atom-project phonon density of states in SI Fig. S4. Such information may be useful for further understanding the relation between the atomic structure and spin lifetime.

“3. On P. 3, the authors state that at low T and low nc , CsPbBr_3 has a relatively long spin lifetime. Similar to TMDs, this phenomenon is attributed to the spin-valley locking, where the strength of the spin-valley locking has been quantitatively analyzed by measuring the spin lifetimes intervalley and intravalley. How about this in CsPbBr_3 ?”

Spin-valley locking (SVL) is irrelevant for CsPbBr_3 .

This is because it means after highly polarized excess spins are generated in one or some valleys of the electronic band structure, they are locked there and can hardly go to the other valley(s) with opposite spin polarization. So SVL can present only if (i) there exist multiple valleys; (ii) spins in each valley are highly polarized and (iii) the spin-flip transitions within each valley are forbidden while spin-flip transitions between different valleys are slow.

Since orthorhombic CsPbBr₃ have only one valley – namely the Γ valley, there is no SVL. For cubic CsPbBr₃, there is also no SVL, since valley spins are not highly polarized and spin-flip transitions within a valley are not forbidden.

SVL can however present in other halide perovskites with persistent spin helix (PSH) where there exist two valleys with highly polarized spins and opposite spin directions.

To avoid confusions, we changed the related sentence from “This is in fact comparable to the hole τ_v of TMDs and their heterostructures[19, 32, 33], known to be ultralong due to spin-valley locking” to “This is in fact comparable to the ultralong hole τ_v of TMDs and their heterostructures[19, 32, 33], $\geq 2 \mu\text{s}$ at $\sim 5 \text{ K}$ ”. SVL is deleted from the sentence.

“4. The authors claim “compute the spin relaxation time (T_1) and ensemble spin dephasing time (T_2^)” in the Abstract, however, the manuscript does not give a clear discussion of spin dephasing time or data to support the statement of “ensemble spin dephasing time (T_2^*)” in Abstract.”*

Thanks for the suggestions. In the revised manuscript, before presenting our theoretical results of T_1 and T_2^* of bulk (or itinerant or delocalized) carriers of CsPbBr₃, we added brief introductions of these parameters and briefly discussed the possible physical mechanisms that limits T_1 and T_2^* of bulk carriers of halide perovskites.

The related sentences are “Historically, two types of τ_v pave the way for optimizing and controlling spin relaxation and dephasing in halide perovskites.” in the last two paragraphs of subsection “Theory” in Sec. “Results and discussions” on the right column of page 2.

“5. I did not find the difference between panels (c) and (e), (d), and (f) in FIG S3. They are the same, please check.”

Thank the referee for pointing out the mistake in Fig. S3. We double checked the figure and put down the correct vibrational patterns of the phonons. Additionally we explained in the caption that the vibrational patterns were plotted with LO-TO splitting.

“6. The order of pictures described in the manuscript is chaotic, the in-text citation of Fig.1 (b) precedes Fig.1 (a).”

We have fixed this issue.

“7. On page 3, how to explain “This is contradictory to the simple assumption frequently employed in previous experimental studies.”

In previous published experimental studies, it was assumed that Fröhlich interaction dominates spin relaxation. However, we found that Fröhlich interaction is unimportant for the EY spin relaxation process in CsPbBr₃. In our original manuscript, we explain the reason for that by the fact that the spin-flip e-ph matrix element of Fröhlich interaction is smaller than those of other e-ph interactions. This explanation is however not enough in-depth.

In the revised manuscript, we further attributed the reason for the miniature role of the spin-independent nature of the long-range Fröhlich interaction. (Note that its spin-dependent part or SOC correction contains a factor of q/c^2 and is short-ranged, so that its spin-dependent part is not special at all compared with other short-range e-ph interactions and considered not a part of Fröhlich interaction.)

The matrix element of Fröhlich interaction between states “1” and “2” approximately takes the simple spin-independent form $iF \cdot o_{12}/q$, where q is the wavevector length, F is a constant, and o_{12} is the wavefunction overlap integral. Since the spin-flip wavefunction overlap is found tiny $\sim 2e-4$ (while spin-conserving wavefunction overlap is of order 1), the spin-flip Fröhlich interaction is rather weak

and unimportant to spin relaxation. Therefore Fröhlich interaction does not contribute to spin relaxation much.

The related added sentences are: “This is because Fröhlich interaction is the long-range part of e-ph interaction and spin-independent, while the spin dependent part of e-ph interaction is short-ranged.” in the fourth paragraph of the subsection “Analysis of spin-phonon relaxation” on the left column of page 5.

“8. On page 4, font error in the first paragraph.”

We assume that the reviewer meant that the sentence “*Fröhlich interaction is unimportant for spin relaxation*” in italic used the wrong font. However, we used the italic on purpose to emphasize the conclusion that Fröhlich interaction is unimportant for spin relaxation (in CsPbBr₃).

Reviewers' Comments:

Reviewer #1:

Remarks to the Author:

The authors followed and discussed the noted questions in a fair and clear manner. Clarifying and improving the manuscript significantly. To repeat, the work remains interesting, working out spin relaxation processes and their strength, highly relevant. However, a few issues remain.

AQ1: Thank you for the magnification. I assume the label 20K in Fig S6 is a typo?

AQ1.2: I find the wording of the roles of impurities a bit confusing. Fig. 5c and 5d shows a good agreement for the situation without impurities. The values without impurity scattering match and the magnetic field dependence, linear without and non linear with impurities matches the experiment. Meaning impurity scattering needs to be absent to explain the experimental findings. Not only a high concentration can be ruled out, from my impression any. Which is indeed an important finding. For sake of clarity of the paper I would place the entire discussion about impurities for this case in the SI and simply state, that they can not explain the experiment. For this case, dephasing of localized carriers might be considered.

Further coming back to Fig. 1c, the stated situation is shown for $n_i=10^{18}\text{cm}^{-3}$ at really high defect concentration. A reader which might not go into detail, may come to the conclusion from the figure that this explains the situation of spin relaxation at low temperatures, but as seen from Fig. 5 impurities are either absent or the interaction much weaker. For clarity I would place it in the SI and rather show a figure as done for e-e scattering, to emphasize that e-i scattering is insignificant for elevated temperatures and too weak (at concentrations $<10^{18}\text{cm}^{-3}$) to damp T_1 at low temperatures.

AQ2: I agree that the spread of g-factor is the relevant parameter of this paragraph. As the results for Fig. 4b, in particular hole g-factor, are neither stable within the different DFT approaches (Fig. S5) nor match the experimental values Ref. 35 ($g_e = +1.69$ to 2.06 ; $g_h = +0.65$ to $+0.85$) and only the g-factor fluctuations relevant for the dephasing time estimate, I would suggest to place the entire g-factor discussion to the SI.

AQ3: ok.

AQ4: Hmm, I will be than very interested in the results for different doping levels. (The references I find unfortunately rather scattered

*Nano Lett. 2015, 15, 1553–1558; Fig3b seems to be in disagreement with the finding. They report a fluence dependence at room temperature while Fig. 1b suggests a flat dependence with carrier concentration at 300K.

* Am. Chem. Soc. 2021, 143, 46, 19438–19445 is about a 2D perovskite film, they show a flat temperature dependence (Fig. 3b) of the relaxation rate and rule out EY mechanism for them (text).

* Nano Lett. 2023, 23, 205–212 is again about 2D perovskite films, Fig 3 c/d with the power dependence associated with the spin generation.

*Nat. Commun. 11, 5665 (2020), shows indeed a strong flux dependence but a rather flat temperature dependence.)

AQ5: ok.

AQ6: ok.

AQ7: ok.

AQ8: I'm suprised that the influence of hyperfine coupling gives a shorter dephasing time for electrons than for holes. In perovskites the hyperfine coupling for holes is much stronger than for electron, as given in methods section 25 to 1.7ueV . E.g. the experimental demonstration Adv. Materials 2022, 34, 2105263 shows an Overhauser field of $\sim 50\text{mT}$ for holes and 5mT for electrons.

AQ9: I was hoping for a theoretical prediction for many materials, like the band gap calculation

presented in Nat. Com. (2019) 10, 2560 Fig. 5. MAPbBr₃ is for sure interesting, though it is known that the A site contribution has a lower effect on the band parameter than the B and X-sites. The discussion is quite short, too short to answer the general claim of giving information on the spin relaxation in all bulk halide perovskites. (for my taste)

A10: fine

Conclusion: The manuscript "how spin relaxes and dephases in bulk halide perovskites" is an extensive work, with a good insight in relaxation mechanisms and explicit calculations for bulk CsPbBr₃ spin dynamics. The given answers improve significantly the manuscript. With great efforts the calculations are tried to back with experimental findings, some with good agreement some with some remaining questions, but this is part of science. I find the title disputable and "how spin relaxes and dephases in bulk CsPbBr₃ perovskite" more justified. I suggest publication after revision.

Reviewer #2:

Remarks to the Author:

All the concerns from the reviewer are mainly addressed. Although the structure-property relationship is still very unclear, I suggest that it can be published due to the significant improvement made by the authors.

Below, we repeat the reviewer's comments in black italic and present our responses point-by-point in blue color.

REVIEWER COMMENTS

Reviewer #1 (Remarks to the Author):

“The authors followed and discussed the noted questions in a fair and clear manner. Clarifying and improving the manuscript significantly. To repeat, the work remains interesting, working out spin relaxation processes and their strength, highly relevant. However, a few issues remain.”

We thank the reviewer's comment on the manuscript's significant improvement.

We revised our manuscript based on the reviewer's comments. The major revisions are:

- (i) We added spin lifetime results of CsSnBr₃, CsPbCl₃ and CsPbI₃ as a function of temperature, in addition to MAPbBr₃ in a new figure Fig. 7 on Page 9 in the revised manuscript.
- (ii) We moved the subfigures concerning impurity effects as well as related discussions and g-factor dependence on electronic structure methods to SI to avoid possible confusions, according to the reviewer's comments.

We address the reviewer's other comments point-by-point below.

“AQ1: Thank you for the magnification. I assume the label 20K in Fig S6 is a typo?”

Yes, it was a typo. We fixed it.

“AQ1.2: I find the wording of the roles of impurities a bit confusing. Fig. 5c and 5d shows a good agreement for the situation without impurities. The values without impurity scattering match and the magnetic field dependence, linear without and non linear with impurities matches the experiment. Meaning impurity scattering needs to be absent to explain the experimental findings. Not only a high concentration can be ruled out, from my impression any. Which is indeed an important finding. For sake of clarity of the paper I would place the entire discussion about impurities for this case in the SI and simply state, that they can not explain the experiment. For this case, dephasing of localized carriers might be considered. Further coming back to Fig. 1c, the stated situation is shown for $n_i=10^{18}\text{cm}^{-3}$ at really high defect

concentration. A reader which might not go into detail, may come to the conclusion from the figure that this explains the situation of spin relaxation at low temperatures, but as seen from Fig. 5 impurities are either absent or the interaction much weaker. For clarity I would place it in the SI and rather show a figure as done for e-e scattering, to emphasize that e-i scattering is insignificant for elevated temperatures and to weak (at concentrations $<10^{18} \text{ cm}^{-3}$) to damp T1 at low temperatures.”

We thank the reviewer for the great suggestions. We moved the subfigures concerning impurity effects and a majority part of discussions to the SI to avoid possible confusions.

The discussions in the main text concerning impurity effects are significantly shortened and we emphasize that impurity scattering should be insignificant in the current study. We note that including impurity scattering helps validate EY mechanism of spin relaxation in this system, which shows increasing extrinsic scattering decreases spin lifetime.

“AQ2: I agree that the spread of g-factor is the relevant parameter of this paragraph. As the results for Fig. 4b, in particular hole g-factor, are neither stable within the different DFT approaches (Fig. S5) nor match the experimental values Ref. 35 ($g_e = +1.69$ to 2.06 ; $g_h = +0.65$ to $+0.85$) and only the g-factor fluctuations relevant for the dephasing time estimate, I would suggest to place the entire g-factor discussion to the SI.”

We thank the reviewers for the suggestion. We moved the discussions related to the absolute values of the g-factors to the SI. We discussed in detail the g factor dependence on electronic structure methods, e.g. DFT exchange-correlation functionals in SI section SV.

We however still kept discussions related to (i) the g-factor fluctuation amplitude “ Δg ” and (ii) the trend of g-factor changes with the state energy in the main text, because the former is critical for spin dephasing time (T_2^*) while the latter is informative and can be verified by the future experimental and theoretical works.

Our discussions emphasized that both “ Δg ” and the trend of g-factor changes as a function of carrier density are not sensitive to the DFT exchange-correlation functional or electronic structure methods.

“AQ3: ok.”

We thank the reviewer for the confirmation.

“AQ4: Hmm, I will be than very interested in the results for different doping levels. (The references I find unfortunately rather scattered

**Nano Lett. 2015, 15, 1553–1558; Fig3b seems to be in disagreement with the finding. They report a fluence dependence at room temperature while Fig. 1b suggests a flat dependence with carrier concentration at 300K.*

** Am. Chem. Soc. 2021, 143, 46, 19438–19445 is about a 2D perovskite film, they show a flat temperature dependence (Fig. 3b) of the relaxation rate and rule out EY mechanism for them (text).*

** Nano Lett. 2023, 23, 205–212 is again about 2D perovskite films, Fig 3 c/d with the power dependence associated with the spin generation.*

**Nat. Commun. 11, 5665 (2020), shows indeed a strong flux dependence but a rather flat temperature dependence.)”*

We agree with the reviewer that the references are indeed scattered and different people may explain the experimental data in different ways. This fact actually indicates the importance of systematic parameter-free theoretical studies and also more future experimental work.

We note that we are preparing a separate detailed study on the doping dependence on spin lifetime and g factors of perovskite halide, which may address this issue in a more systematic manner later.

AQ5: ok.

We thank the reviewer for the confirmation.

AQ6: ok.

We thank the reviewer for the confirmation.

AQ7: ok.

We thank the reviewer for the confirmation.

“AQ8: I'm suprised that the influence of hyperfine coupling gives a shorter dephasing time for electrons than for holes. In perovskites the hyperfine coupling for holes is much stronger than for electron, as given in methods section 25 to 1.7ueV. E.g. the experimental demonstration Adv. Materials 2022, 34, 2105263 shows an Overhauser field of ~50mT for holes and 5mT for electrons.”

We thank the reviewer to point it out. This is a typo. We have fixed it in the revised manuscript on page 9, right column, last paragraph.

Note that we have written that the constant $C^{(bc)}$ of electrons is smaller than that of holes. Since spin dephasing rate $1/T_2^*$ is approximate to $C^{(bc)}$ / volume of localized carriers; with the same volume, spin dephasing time T_2^* of electrons should be longer than the one of holes.

“AQ9: I was hoping for a theoretical prediction for many materials, like the band gap calculation presented in Nat. Com. (2019) 10, 2560 Fig. 5. MAPbBr₃ is for sure interesting, though it is known that the A side contribution has a lower effect on the band parameter than the B and X-sites. The discussion is quite short, too short to answer the general claim of giving information on the spin relaxation in all bulk halide perovskites. (for my taste)”

We thank the reviewer for the great suggestions. We computed additional 3 systems and added a new figure Fig. 7 showing spin lifetime of MAPbBr₃, CsSnBr₃, CsPbCl₃ and CsPbI₃, compared with CsPbBr₃, and the related detailed discussions can be found in the revised manuscript page 9, left column, second paragraph, “As an initial study...”

Our theoretical results indicate that chemical composition effects are not very strong, and usually change spin lifetimes by tens of percent or a few times in a wide temperature range. Our results also show a clear trend for CsPbX₃ spin lifetime (while keeping the same crystal symmetry): CsPbCl₃ > CsPbBr₃ > CsPbI₃. We note additional spin lifetime study including symmetry and dimensionality will be carried out in the future.

A10: fine

We thank the reviewer for the confirmation.

“Conclusion: The manuscript "how spin relaxes and dephases in bulk halide perovskites" is an extensive work, with a good insight in relaxation mechanisms and explicit calculations for bulk CsPbBr₃ spin dynamics. The given answers improve significantly the manuscript. With great efforts the calculations are tried to back with experimental findings, some with good agreement some with some remaining questions, but this is part of science. I find the title disputable and "how spin relaxes and dephases in bulk CsPbBr₃ perovskite" more justified. I suggest publication after revision.”

We thank the reviewer for the great suggestions, and we believe our revised manuscript has addressed the reviewers' questions.

Reviewer #2 (Remarks to the Author):

“All the concerns from the reviewer are mainly addressed. Although the structure-property relationship is still very unclear, I suggest that it can be published due to the significant improvement made by the authors.”

We thank the reviewer for the positive comments to our work. We believe after adding the new systems for understanding the chemical composition effects, we have addressed the reviewer’s concerns. We note we are performing future work on symmetry and dimensionality effect on spin lifetime.

Reviewers' Comments:

Reviewer #1:

Remarks to the Author:

The authors addressed my concerns and resolved them satisfactorily. The manuscript is now consistent and ready for publication.